# PATCHCODE: Discrete Latent Predictive Learning for EEG Foundation Model

Kieren Yu [1]   Ziyang Liu [1]   Chang Huang [1]   Kaishun Wu [1]

## Abstract

EEG foundation models aim to learn transferable representations, yet EEG recordings are dominated by high-frequency noise and large cross-subject variability. Existing pretraining strategies such as masked autoencoding or autoregressive modeling often treat waveform reconstruction as the learning signal, making the objective sensitive to stochastic fluctuations rather than consistent neurophysiological structure. To address this overlap, we propose **PATCHCODE**, a region-aware discrete predictive learning framework that keeps the encoder input continuous while introducing region-aware discrete codes as stable supervision targets. We pretrain a masked predictive encoder on continuous EEG patches with dual-granularity learning: it predicts missing patch-level representations to preserve fine spatiotemporal structure, while aligning them to discretized code targets from a frozen tokenizer to anchor robust semantics. Extensive experiments across sixteen downstream datasets spanning emotion recognition, motor imagery, sleep staging, seizure detection, vigilance estimation, stress detection, and clinical diagnosis demonstrate that PATCHCODE achieves competitive performance compared to state-of-the-art baselines, with notable gains in data efficiency under limited labels. Our code is available at github.com/kierenyu/Patchcode.

## 1. Introduction

The paradigm of training foundation models on vast corpora of unlabeled data has advanced natural language processing (Brown et al., 2020; Devlin et al., 2019) and computer vision (Dosovitskiy et al., 2021; He et al., 2022). By leveraging self-supervised learning (SSL), these models acquire generalizable representations that transfer effectively to downstream tasks. Recently, this success has motivated efforts to develop EEG foundation models (EEG-FMs) that can learn universal representations from large-scale brain signal data (Jiang et al., 2024; Yang et al., 2023; Wang et al., 2025a). Unlike text or images, EEG signals present distinct challenges: low signal-to-noise ratios (SNR), substantial cross-subject variability arising from individual neural anatomy and recording conditions, and complex temporal-spectral dynamics spanning multiple frequency bands (Kostas et al., 2021). These characteristics demand SSL objectives specifically designed for noisy, stochastic neural signals.

Recent EEG-FMs have explored diverse pre-training paradigms. *Reconstruction-based methods* adopt masked autoencoding to recover masked EEG segments. LaBraM (Jiang et al., 2024) pre-trains on 2,500 hours of EEG data using neural tokenization and masked prediction, while Brain-Wave (Yuan et al., 2024) scales to more than 40,000 hours with a foundation-model objective for clinical brain signals. CBraMod (Wang et al., 2025a) introduces criss-cross attention to capture both temporal and spatial dependencies during reconstruction. However, waveform-level reconstruction may encourage models to fit high-frequency noise rather than abstract meaningful neural dynamics. *Contrastive methods* learn representations by maximizing agreement between augmented views. BENDR (Kostas et al., 2021) applies contrastive learning with temporal neighborhood coding, and BIOT (Yang et al., 2023) extends this to multi-modal biosignals. Yet designing semantics-preserving augmentations for EEG remains challenging, as standard transformations can alter physiological content. *Autoregressive methods* treat EEG as sequential tokens for next-token prediction. EEGPT (Wang et al., 2024) and NeuroLM (Jiang et al., 2025) adopt GPT-style objectives, while THD-BAR (Yang et al., 2026) incorporates topological hierarchy. *Discrete tokenization* has emerged as a promising direction: DeWave (Duan et al., 2023) pioneers discrete encoding for brain-to-text, and CodeBrain (Ma et al., 2026) explores decoupled tokenizer architectures. However, existing methods typically use discrete codes as encoder inputs or reconstruction targets, rather than as abstract supervision signals for latent prediction.

**Discrete supervision and predictive SSL.** Predicting discrete targets has been effective in speech and vision SSL (e.g., HuBERT (Hsu et al., 2021), wav2vec 2.0 (Baevski

[1]The Hong Kong University of Science and Technology (Guangzhou), Guangzhou, China. Correspondence to: Kaishun Wu <wuks@hkust-gz.edu.cn>.

*Proceedings of the 43rd International Conference on Machine Learning*, Seoul, South Korea. PMLR 306, 2026. Copyright 2026 by the author(s).

et al., 2020), BEiT (Bao et al., 2022), iBOT (Zhou et al., 2022)), where continuous inputs are regularized by noise-robust categorical constraints. Complementary to this, *Joint-Embedding Predictive Architectures (JEPA; Joint-Embedding Predictive Architecture)* (LeCun et al., 2022; Assran et al., 2023) learn by predicting latent embeddings of masked regions, avoiding pixel-/waveform-level reconstruction. In practice, JEPA-style training is often instantiated via teacher–student predictors with momentum targets (Grill et al., 2020; Caron et al., 2021; Baevski et al., 2022) to stabilize learning. We adapt these predictive ideas to EEG, where low SNR and cross-subject variability motivate combining continuous inputs with discrete code supervision.

Motivated by these observations, we explore an alternative paradigm: *discrete latent predictive learning*. Rather than reconstructing raw waveforms, the core encoder learns to predict abstract representations in a latent space where stochastic noise has been filtered. To this end, we introduce **PATCHCODE**, a two-stage framework that decouples the discovery of discrete codes (Stage A) from predictive representation learning (Stage B).

PATCHCODE operates through a **decoupled training procedure**. In Stage A, we train a Region-aware Tokenizer that maps continuous EEG patches to discrete code indices via vector quantization, supervised by frequency-domain reconstruction (predicting FFT amplitude and phase) and region-based contrastive regularization. Importantly, after Stage A training, the tokenizer is frozen and its decoder is discarded. In Stage B, we employ a Student-Teacher architecture with Region-Bias Transformer encoders, trained through *JEPA-style* (Joint-Embedding Predictive Architecture) objectives (Assran et al., 2023; LeCun et al., 2022): the encoder processes *continuous*, masked EEG patches while being optimized through three complementary objectives—Patch-to-Patch (P2P), Patch-to-Code (P2C), and Code-to-Patch (C2P). Unlike standard JEPA that only predicts continuous representations, our framework also incorporates discrete code supervision from the frozen tokenizer. The discrete codes serve purely as supervision targets—they are *never* fed into the encoder as inputs. This design enables the encoder to retain fine-grained temporal information from continuous inputs while being regularized by noise-filtered discrete targets.

Our main contributions are summarized as follows:

- **Two-Stage Decoupled Framework:** We propose PATCHCODE, which separates discrete token learning (Stage A, with spectral supervision) from representation learning (Stage B, with JEPA-style predictive objectives), allowing the encoder to process continuous signals while receiving discrete supervision.

- **Multi-Pathway Predictive Objectives:** We design three complementary prediction pathways (P2P, P2C, C2P) that jointly enforce continuous and discrete representation consistency. This multi-pathway design provides task-adaptive inductive biases: P2P captures fine-grained dynamics, P2C enforces categorical constraints, and C2P regularizes the Student encoder by imposing structural constraints from the discrete code space.

- **Empirical Evaluation:** Experiments on sixteen downstream datasets across seven EEG domains show that PATCHCODE achieves competitive or improved performance compared to recent baselines, with substantial gains under limited labeled data scenarios. Ablation studies confirm complementary contributions of components across different task types.

## 2. Method

We introduce PATCHCODE, a two-stage framework that decouples *discrete code discovery* (Stage A) from *predictive representation learning* (Stage B). Unlike generative approaches that reconstruct raw waveforms, our paradigm learns robust EEG representations by predicting patch-level features under discrete code supervision from a frozen tokenizer. This design enables the encoder to process fine-grained continuous signals while being regularized by abstract discrete targets, effectively filtering stochastic noise without sacrificing temporal resolution.

### 2.1. Unified Topological Mapping and Patching

**Input Formulation.** Given a raw EEG recording $\mathbf{X} \in \mathbb{R}^{B \times C \times T}$ with batch size $B$, $C$ channels, and $T$ time samples, we align all recordings to a canonical 21-channel configuration based on the international 10-20 system (HH, 1958) via spherical spline interpolation (Perrin et al., 1989). This ensures cross-dataset compatibility while preserving neuroanatomical topology. All signals are bandpass filtered (0.5–45 Hz) (Widmann et al., 2015) and resampled to 200 Hz.

**Patching.** We segment the temporal dimension into $N$ non-overlapping patches of length $P$ samples (default $P$=200 at 200 Hz, corresponding to 1-second windows). The patched input becomes $\mathbf{X}_p \in \mathbb{R}^{B \times C \times N \times P}$. We flatten the channel and temporal dimensions to obtain a sequence of $L = C \times N$ patch tokens, where each token $\mathbf{x}_i \in \mathbb{R}^P$ represents a single-channel, single-window EEG segment.

**Token Embedding.** Each patch token is first projected via a linear layer to dimension $D$. We then inject structural information through an additive embedding strategy:

$$\mathbf{E}_{in} = \text{Linear}(\mathbf{X}_p) + \mathbf{E}_{chan} + \mathbf{E}_{time} + \mathbf{E}_{region} \quad (1)$$

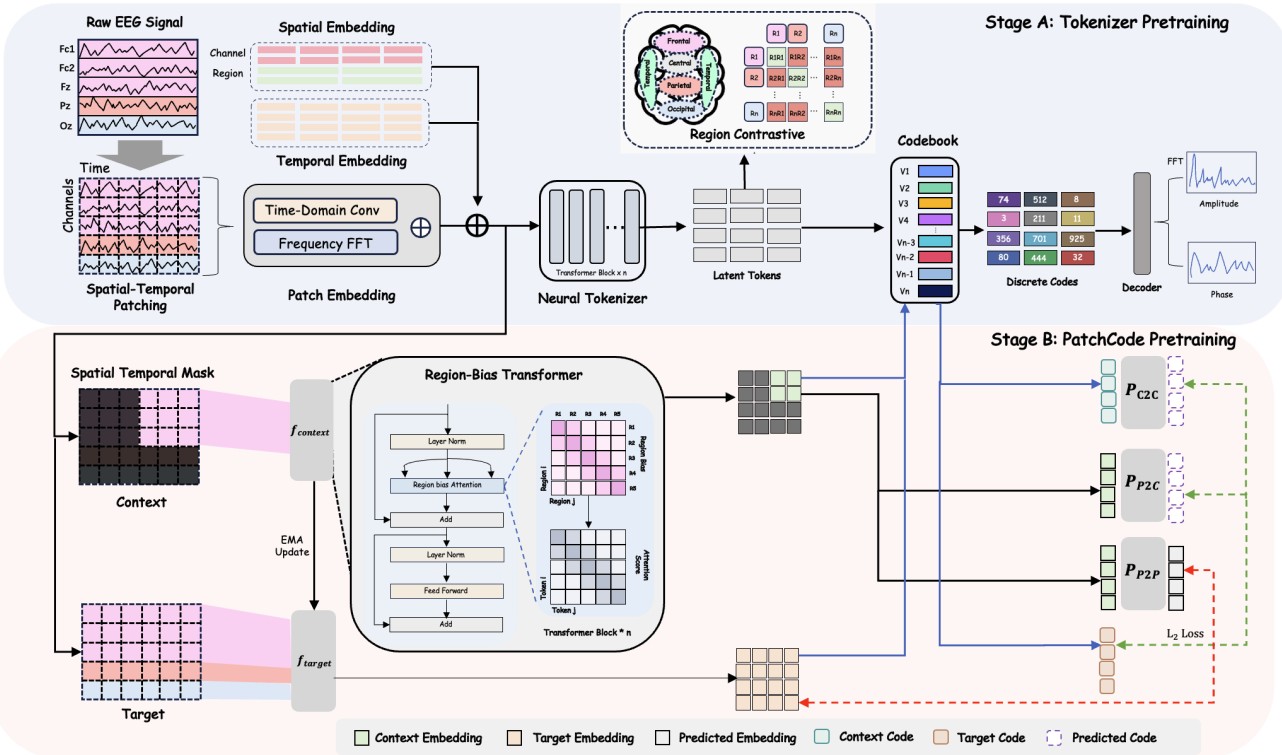

*Figure 1.* **Overview of PATCHCODE.** The framework consists of two decoupled stages. **Stage A** trains a Region-Aware Tokenizer that maps continuous EEG patches to discrete codes via vector quantization, supervised by spectral reconstruction and region contrastive objectives. **Stage B** employs a Student-Teacher architecture with Region-Bias Transformer encoders, trained through JEPA-style predictive objectives. The Student processes masked continuous patches and learns through three complementary objectives: P2P (Patch-to-Patch), P2C (Patch-to-Code), and C2P (Code-to-Patch). The frozen tokenizer from Stage A provides discrete code supervision without entering the encoder pathway.

where $\mathbf{E}_{chan} \in \mathbb{R}^{C \times D}$ encodes channel identity, $\mathbf{E}_{time} \in \mathbb{R}^{N \times D}$ encodes temporal position, and $\mathbf{E}_{region} \in \mathbb{R}^{R \times D}$ encodes anatomical region membership (e.g., Frontal, Central, Parietal, Temporal, Occipital). Each token inherits embeddings according to its channel-time-region assignment, yielding the final input $\mathbf{E}_{in} \in \mathbb{R}^{B \times L \times D}$.

## 2.2. Stage A: Region-Aware Tokenizer

Stage A trains a tokenizer $\mathcal{T}$ that converts continuous EEG patches into discrete codes via Vector Quantization (VQ) (Van Den Oord et al., 2017). The tokenizer encoder $\text{Enc}_A$ maps each patch to a continuous latent vector $\mathbf{z}_i \in \mathbb{R}^D$, which is quantized to the nearest entry in a learnable codebook $\mathcal{C} = \{\mathbf{e}_k\}_{k=1}^K$:

$$q_i = \arg \min_{k \in \{1,\ldots,K\}} \|\mathbf{z}_i - \mathbf{e}_k\|_2, \quad \mathbf{z}_i^q = \mathbf{e}_{q_i} \quad (2)$$

where $q_i$ is the discrete code index. The codebook is updated via EMA (Razavi et al., 2019; Dhariwal et al., 2020) to prevent collapse.

To avoid fitting time-domain noise, we supervise the tokenizer with **frequency-domain targets**: the decoder pre-

dicts FFT amplitude and phase of the first $P_f$ frequency bins. The training objective combines:

$$\mathcal{L}_A = \mathcal{L}_{vq} + \lambda_{spec}\mathcal{L}_{spec} + \lambda_{reg}\mathcal{L}_{reg} - \lambda_{usage}\mathcal{L}_{usage} \quad (3)$$

where $\mathcal{L}_{vq}$ is the VQ commitment/codebook loss (with straight-through gradients), $\mathcal{L}_{spec}$ matches predicted vs. target FFT amplitude/phase, $\mathcal{L}_{reg}$ is a region-wise contrastive regularizer that pulls together token embeddings within the same anatomical region, and $\mathcal{L}_{usage}$ maximizes code-usage entropy to prevent collapse. After training, we freeze the tokenizer and discard the decoder; full definitions and training details are in Appendix B.

## 2.3. Stage B: PatchCode Pre-training

Stage B trains the main representation encoder through JEPA-style predictive objectives (Assran et al., 2023; LeCun et al., 2022). We employ a Student-Teacher architecture (Grill et al., 2020; Caron et al., 2021) with Region-Bias Transformer encoders, where the Student processes masked tokens and predicts the Teacher's representations at masked positions. Unlike standard JEPA that predicts only continuous representations, our framework extends this paradigm

**Algorithm 1** Stage A: Region-Aware Tokenizer Training

---

**Input:** EEG dataset $\mathcal{D}$, codebook size $K$, region map $\mathcal{R}$
**Initialize:** Encoder $\text{Enc}_A$, Decoder $\text{Dec}_A$, Codebook $\mathcal{C} = \{\mathbf{e}_k\}_{k=1}^{K}$
**for** each minibatch $\mathbf{X} \in \mathcal{D}$ **do**
    1. Patch and embed: $\mathbf{E}_{in} \leftarrow \text{Embed}(\mathbf{X})$
    2. Encode: $\mathbf{Z} = \text{Enc}_A(\mathbf{E}_{in})$        $\triangleright \mathbf{Z} \in \mathbb{R}^{B \times L \times D}$
    3. Quantize: $q_i = \arg\min_k \|\mathbf{z}_i - \mathbf{e}_k\|_2$, $\mathbf{Z}^q = \mathcal{C}[q]$
    4. Compute spectral targets: $\mathbf{A}, \boldsymbol{\Phi} \leftarrow \text{FFT}(\mathbf{X})$
    5. Decode: $\widehat{\mathbf{A}}, \widehat{\boldsymbol{\Phi}} = \text{Dec}_A(\mathbf{Z}^q)$
    6. Compute $\mathcal{L}_A = \mathcal{L}_{vq} + \lambda_{spec}\mathcal{L}_{spec} + \lambda_{reg}\mathcal{L}_{reg} - \lambda_{usage}\mathcal{L}_{usage}$
    7. Update $\text{Enc}_A, \text{Dec}_A$ via $\nabla\mathcal{L}_A$; Update $\mathcal{C}$ via EMA
**end for**
**Output:** Frozen Tokenizer $\mathcal{T} = (\text{Enc}_A, \mathcal{C})$    $\triangleright$ Decoder discarded

---

by incorporating discrete code supervision from the frozen Stage A tokenizer—analogous to how HuBERT (Hsu et al., 2021) and BEiT (Bao et al., 2022) leverage discrete targets in speech and vision. The key design principle is that the encoder **always processes continuous EEG patches**, while discrete codes serve purely as supervisory signals—they never enter the encoder pathway. This decoupling allows the encoder to retain fine-grained temporal information while being regularized by noise-filtered discrete targets.

**Continuous Token Construction.** Unlike Stage A which produces discrete codes, Stage B constructs continuous token representations directly from raw EEG. For each patch $\mathbf{x}_i \in \mathbb{R}^P$, we compute dual-pathway embeddings: (1) a *temporal pathway* via linear projection $\mathbf{z}_i^{temp} = \text{Linear}(\mathbf{x}_i) \in \mathbb{R}^{D/2}$, and (2) a *spectral pathway* via FFT amplitude $\mathbf{z}_i^{spec} = \text{Linear}(|\text{FFT}(\mathbf{x}_i)|) \in \mathbb{R}^{D/2}$. These are concatenated and fused: $\mathbf{z}_i = \text{Fusion}([\mathbf{z}_i^{temp}; \mathbf{z}_i^{spec}])$. The final token embedding incorporates three positional encodings:

$$\mathbf{E}_{in} = \text{Proj}(\mathbf{z}) + \mathbf{E}_{time} + \mathbf{E}_{region} + \mathbf{E}_{chan} \quad (4)$$

**Region-Bias Transformer Encoder.** The encoder $f_\theta$ is a Transformer with *region-aware attention bias*. We maintain a learnable bias table $\mathbf{B} \in \mathbb{R}^{R \times R}$ where $R$ is the number of brain regions. For tokens $i, j$ belonging to regions $r_i, r_j$ respectively, the attention logits are modified as:

$$\text{Attn}(Q, K, V) = \text{softmax}\left(\frac{QK^\top}{\sqrt{d}} + \mathbf{B}[r_i, r_j]\right)V \quad (5)$$

where $\mathbf{B}[r_i, r_j]$ is broadcast across all attention heads. This bias encodes learnable priors about inter-region interactions (e.g., stronger connectivity within frontal regions for emotion tasks). Each Transformer block follows a Pre-LN archi-

tecture: LayerNorm $\rightarrow$ Region-Bias Attention $\rightarrow$ Residual $\rightarrow$ LayerNorm $\rightarrow$ FFN $\rightarrow$ Residual.

We emphasize that region bias is a spatial inductive bias rather than the primary noise-filtering mechanism. The frequency-domain tokenizer and discrete bottleneck provide the direct noise-robust supervision, while region-biased attention separately encourages representations to respect known EEG topography and reduces reliance on spatially unstructured artifacts.

**Student-Teacher Architecture.** We employ an asymmetric Student-Teacher framework (Grill et al., 2020; Caron et al., 2021). The **Student** $f_\theta$ processes *masked* tokens where target positions are replaced with a learnable [MASK] embedding. The **Teacher** $f_{\bar{\theta}}$ processes *unmasked* tokens and is updated via exponential moving average (EMA): $\bar{\theta} \leftarrow \tau\bar{\theta} + (1 - \tau)\theta$, where $\tau$ is the momentum coefficient (default 0.996). This EMA update provides stable, slowly-evolving targets that prevent representational collapse (Chen & He, 2021), a common failure mode in self-supervised learning without negative samples. Both encoders share the same Region-Bias Transformer architecture but with decoupled weights.

**Structured Block Masking.** We apply JEPA-style context-target masking on the spatio-temporal token grid. Specifically, we sample contiguous *context blocks* (spanning 1–2 regions $\times$ consecutive time steps) and treat remaining tokens as *targets* to predict. Context tokens retain their embeddings while target positions receive the [MASK] token. This strategy forces the model to predict across temporal and spatial gaps, encouraging long-range dependency learning. Let $\mathcal{M} \subset \{1, \ldots, L\}$ denote the set of masked (target) token indices.

**Code Index Generation.** For each input, the frozen Stage A tokenizer produces code indices $\mathbf{q} = \{q_i\}_{i=1}^{L}$ where $q_i \in \{1, \ldots, K\}$. Crucially, these codes are computed from the **original unmasked input** and serve purely as super-vision targets—*they do not enter the Student or Teacher encoder pathways*. We also retrieve the corresponding code embeddings $\mathbf{e}_{q_i}$ from the frozen codebook for the C2P objective.

**Objective 1: Patch-to-Patch (P2P).** The P2P loss trains the Student to predict the Teacher's patch representations at masked positions. This encourages the Student to learn contextually-informed features that match the Teacher's view of the complete input:

$$\mathcal{L}_{P2P} = \frac{1}{|\mathcal{M}|} \sum_{i \in \mathcal{M}} \left\|\text{Pred}_{P2P}(\mathbf{h}_i^s) - \text{sg}(\mathbf{h}_i^t)\right\|_2^2 \quad (6)$$

where $\text{Pred}_{P2P}$ is a 2-layer MLP predictor and $\text{sg}(\cdot)$ denotes stop-gradient on the Teacher output.

**Objective 2: Patch-to-Code (P2C).** The P2C loss is the **core discrete supervision signal**. It trains the Student to predict the code index $q_i$ from its patch representation at masked positions via a classification head:

$$\mathcal{L}_{P2C} = \frac{1}{|\mathcal{M}|} \sum_{i \in \mathcal{M}} \text{CrossEntropy}\big(\text{Cls}_{P2C}(\mathbf{h}_i^s), \, q_i\big) \quad (7)$$

where $\text{Cls}_{P2C} : \mathbb{R}^D \to \mathbb{R}^K$ is a linear classifier. The target $q_i$ comes from the frozen tokenizer applied to the original input. This objective provides a noise-robust learning signal: since the tokenizer was trained with spectral objectives, the codes abstract away high-frequency stochastic variations while preserving neurophysiologically meaningful patterns.

**Objective 3: Code-to-Patch (C2P).** The C2P loss provides a complementary pathway by predicting the Student's patch representation from the discrete code embedding. For each masked position $i$, we retrieve the code embedding $\mathbf{e}_{q_i}$ from the frozen Stage A codebook and predict the Student's patch feature:

$$\mathcal{L}_{C2P} = \frac{1}{|\mathcal{M}|} \sum_{i \in \mathcal{M}} \big\| \text{Pred}_{C2P}(\mathbf{e}_{q_i}) - \mathbf{h}_i^s \big\|_2^2 \quad (8)$$

where $\text{Pred}_{C2P}$ is a separate 2-layer MLP, $\mathbf{e}_{q_i}$ is frozen (no gradient to codebook), but $\mathbf{h}_i^s$ is *not* stop-gradiented. **Note on gradient flow:** Unlike a teacher-targeted variant, we formulate C2P as predicting the Student patch representation from the frozen code embedding. This allows gradients from $\mathcal{L}_{C2P}$ to update the Student encoder $f_\theta$ and region bias table $\mathbf{B}$, making C2P an effective regularizer that enforces compatibility between the discrete code space and the continuous representation space. The Teacher branch remains stop-gradiented for P2P stability, while C2P acts as a structural constraint that improves Student representation quality and downstream transfer.

**Stage B Total Objective.** The complete Stage B loss combines all three terms:

$$\mathcal{L}_B = \lambda_{P2P}\mathcal{L}_{P2P} + \lambda_{P2C}\mathcal{L}_{P2C} + \lambda_{C2P}\mathcal{L}_{C2P} \quad (9)$$

where $\lambda_{P2C}$ is set as the primary weight, while $\lambda_{P2P}$ and $\lambda_{C2P}$ serve as auxiliary stabilizers. In the final configuration, we use $\lambda_{P2P} = 1.0$, $\lambda_{P2C} = 1.5$, and $\lambda_{C2P} = 0.3$, selected by validation performance and the sensitivity analysis in Appendix D.4. We optionally apply curriculum scheduling to $\lambda_{P2C}$, using higher weight early in training to establish discrete grounding before gradually balancing with continuous objectives.

---

**Algorithm 2** Stage B: PatchCode Pre-training

---

**Input:** EEG dataset $\mathcal{D}$, Frozen Tokenizer $\mathcal{T}$, EMA momentum $\tau$
**Initialize:** Student $f_\theta$, Teacher $f_{\bar{\theta}} \leftarrow f_\theta$
       Predictors: $\text{Pred}_{P2P}$, $\text{Cls}_{P2C}$, $\text{Pred}_{C2P}$
       Region bias table: $\mathbf{B} \in \mathbb{R}^{R \times R}$
**for** each minibatch $\mathbf{X} \in \mathcal{D}$ **do**
    *// Discrete supervision from frozen Stage A*
    $\mathbf{q}, \mathbf{E}^{code} \leftarrow \mathcal{T}(\mathbf{X})$           ▷ no gradient
    *// Continuous token construction*
    $\mathbf{E}_{in} \leftarrow \text{PatchEmbed}(\mathbf{X}) + \mathbf{E}_{time} + \mathbf{E}_{region} + \mathbf{E}_{chan}$
    *// JEPA-style masking*
    $\mathcal{M} \leftarrow \text{SampleContextTargetMask}()$
    $\tilde{\mathbf{E}}_{in} \leftarrow \text{ApplyMask}(\mathbf{E}_{in}, \mathcal{M})$
    *// Region-bias attention*
    $\mathbf{A}_{bias} \leftarrow \mathbf{B}[\mathbf{r}_i, \mathbf{r}_j]$     ▷ lookup by region ids
    *// Forward pass*
    $\mathbf{H}^s \leftarrow f_\theta(\tilde{\mathbf{E}}_{in}, \mathbf{A}_{bias})$       ▷ Student
    $\mathbf{H}^t \leftarrow f_{\bar{\theta}}(\mathbf{E}_{in}, \mathbf{A}_{bias})$    ▷ Teacher (stop-grad)
    *// Compute losses on masked positions*
    $\mathcal{L}_{P2P} \leftarrow \text{MSE}(\text{Pred}_{P2P}(\mathbf{H}_{\mathcal{M}}^s), \text{sg}(\mathbf{H}_{\mathcal{M}}^t))$ ▷ Teacher stop-grad
    $\mathcal{L}_{P2C} \leftarrow \text{CE}(\text{Cls}_{P2C}(\mathbf{H}_{\mathcal{M}}^s), \mathbf{q}_{\mathcal{M}})$
    $\mathcal{L}_{C2P} \leftarrow \text{MSE}(\text{Pred}_{C2P}(\mathbf{E}_{\mathcal{M}}^{code}), \mathbf{H}_{\mathcal{M}}^s)$ ▷ Student *not* stop-grad
    $\mathcal{L}_B \leftarrow \lambda_{P2P}\mathcal{L}_{P2P} + \lambda_{P2C}\mathcal{L}_{P2C} + \lambda_{C2P}\mathcal{L}_{C2P}$
    *// Update: all three losses backprop to Student encoder*
    Update $\theta$ (Student encoder), predictors, $\mathbf{B}$ via $\nabla\mathcal{L}_B$
    $\bar{\theta} \leftarrow \tau\bar{\theta} + (1 - \tau)\theta$     ▷ Teacher EMA (no grad)
**end for**
**Output:** Pre-trained Encoder $f_\theta$

---

## 3. Experiments

### 3.1. Experimental Setup

#### 3.1.1. PRE-TRAINING CONFIGURATION

We pre-train PATCHCODE on the Temple University Hospital EEG Corpus (TUH-EEG v2.0.0) (Obeid & Picone, 2016), which contains over 30,000 hours of clinical EEG data from 25,000 recording sessions. All recordings are resampled to 200 Hz and preprocessed with a 0.5–45 Hz bandpass filter. We train three model variants: Tiny (2M parameters), Base (9M parameters), and Large (37M parameters), using the AdamW optimizer with cosine learning rate scheduling. Training is conducted on 4× NVIDIA A100 GPUs for 300 epochs. Complete pre-training configuration details, including architecture specifications, loss weights, and optimization hyperparameters, are provided in Appendix B and Appendix D.

*Table 1.* Overview of Downstream EEG datasets and tasks used for evaluation.

| Task | Dataset | Rate (Hz) | #Channels | Duration | #Samples | #Subjects | Label |
|------|---------|-----------|-----------|----------|----------|-----------|-------|
| Emotion Recognition | FACED | 250 | 32 | 10s | 10,332 | 123 | 9-class |
| | SEED-V | 200 | 62 | 1s | 117,744 | 16 | 5-class |
| Motor Imagery | BCI-IV-2a | 250 | 22 | 4s | 11,988 | 9 | 4-class |
| | PhysioNet-MI | 160 | 64 | 4s | 9,837 | 109 | 4-class |
| Sleep Staging | ISRUC (S1) | 200 | 6 | 30s | 87,187 | 100 | 5-class |
| | HMC | 256 | 16 | 30s | 326,993 | 154 | 5-class |
| Seizure Detection | CHB-MIT | 256 | 23 | 1s | 6,602 | 23 | 2-class |
| | Siena | 512 | 29 | 1s | 58,203 | 14 | 2-class |
| Event Type Classification | TUEV | 250 | 16 | 5s | 112,491 | 370 | 6-class |
| Abnormal Detection | TUAB | 250 | 16 | 10s | 409,455 | 2,383 | 2-class |

### 3.1.2. DOWNSTREAM DATASETS

We evaluate PATCHCODE on ten primary downstream datasets spanning five primary EEG application domains: emotion recognition (SEED-V (Zheng & Lu, 2015), FACED (Chen et al., 2023)), motor imagery (BCI-IV-2a (Tangermann et al., 2012), PhysioNet-MI (Goldberger et al., 2000; Schalk et al., 2004)), sleep staging (HMC (Alvarez-Estevez & Rijsman, 2021), ISRUC (Khalighi et al., 2016)), seizure detection (CHB-MIT (Shoeb, 2009), Siena (Detti et al., 2020)), and clinical EEG analysis (TUAB (Lopez et al., 2015), TUEV (Harati et al., 2014)). Table 1 summarizes the primary dataset characteristics. Detailed downstream results across all sixteen datasets are provided in Appendix G.

### 3.1.3. BASELINE SELECTION

We compare PATCHCODE against two categories of baselines: (1) **EEG foundation models** pre-trained on large-scale unlabeled data, including BIOT (Yang et al., 2023), LaBraM (Jiang et al., 2024), CBraMod (Wang et al., 2025a), and CSBrain (Zhou et al., 2026); and (2) **supervised baselines** trained from scratch, including EEGNet (Lawhern et al., 2018), EEGConformer (Song et al., 2022), DeepConvNet/ShallowConvNet (Schirrmeister et al., 2017), FBCNet (Mane et al., 2021), and task-specific architectures such as DeepSleepNet (Supratak et al., 2017) and SeqSleepNet (Phan et al., 2019). Detailed descriptions of all baselines are provided in Appendix E.

### 3.1.4. EVALUATION PROTOCOL AND BASELINE FAIRNESS

To ensure fair comparison, we standardize downstream adaptation, preprocessing, and data splits across all methods. Following common EEG-FM evaluation practice (Jiang et al., 2024; Wang et al., 2025a; Zhou et al., 2026), we freeze pre-trained encoders and train a linear head, while supervised baselines are trained end-to-end under the same tuning budget. All inputs are processed with the same resampling/bandpass pipeline and mapped to a canonical 21-channel 10-20 montage (Section 2.1). For EEG-FM base-lines, we re-implement and pre-train on the same 30,000-hour TUH-EEG corpus to remove confounds from pre-training data scale and preprocessing. We verify subject-disjointness between TUH pre-training and TUH-family downstream benchmarks (TUAB/TUEV) to prevent leakage. Hyperparameters are selected using a matched validation protocol and fixed search budget across methods. We report mean ± std over 5 seeds; full protocol details are provided in Appendix F.

### 3.1.5. EVALUATION METRICS

We employ multiple complementary metrics tailored to the characteristics of each task. For multi-class classification (emotion, motor imagery, sleep staging), we report Balanced Accuracy, Cohen's Kappa, and Weighted F1-Score to account for class imbalance. For binary classification with severe class imbalance (seizure detection, abnormal detection), we also report AUC-ROC and AUC-PR. We use standard metrics appropriate for each task type: **Balanced Accuracy** and **Cohen's Kappa** for imbalanced classification, **AUC-PR** for rare event detection.

### 3.2. Main Results

Table 2 presents the performance comparison across four representative downstream tasks. We highlight several key observations:

**Consistent improvements across tasks.** PATCHCODE demonstrates competitive or improved performance across all evaluated tasks. On SEED-V emotion recognition, PATCHCODE (Large) achieves 43.46% balanced accuracy, outperforming the best baseline CSBrain (41.97%) by 1.49%. For BCI-IV-2a motor imagery, PATCHCODE (Large) reaches 58.42% balanced accuracy compared to CS-Brain's 56.57%. On HMC sleep staging, our model achieves 75.56% balanced accuracy with Cohen's $\kappa = 0.705$, indicating substantial agreement. For CHB-MIT seizure detection, PATCHCODE attains 90.85% AUC-ROC, demonstrating strong discriminative ability on imbalanced clinical data.

**Foundation models outperform supervised baselines.** All

*Table 2.* Performance comparison of EEG foundation models on downstream tasks. PATCHCODE demonstrates competitive performance and several best results across tasks.

| METHOD | SEED-V (EMOTION) | | | BCI-IV-2A (MOTOR IMAGERY) | | |
|---|---|---|---|---|---|---|
| | BAL. ACC. | COHEN'S $\kappa$ | W. F1 | BAL. ACC. | COHEN'S $\kappa$ | W. F1 |
| EEGNET | 0.2961±0.0102 | 0.1006±0.0143 | 0.2749±0.0098 | 0.4462±0.0113 | 0.2647±0.0109 | 0.4302±0.0124 |
| EEGCONFORMER | 0.3537±0.0112 | 0.1772±0.0174 | 0.3487±0.0136 | 0.4677±0.0148 | 0.2917±0.0175 | 0.4582±0.0139 |
| SPARCNET | 0.2949±0.0078 | 0.1121±0.0139 | 0.2979±0.0083 | 0.4697±0.0141 | 0.2871±0.0129 | 0.4460±0.0137 |
| CONTRAWR | 0.3546±0.0105 | 0.1905±0.0188 | 0.3544±0.0121 | 0.4682±0.0137 | 0.2894±0.0141 | 0.4434±0.0153 |
| CNN-TRANSFORMER | 0.3678±0.0078 | 0.2072±0.0183 | 0.3642±0.0088 | 0.4583±0.0161 | 0.2825±0.0139 | 0.4473±0.0128 |
| FFCL | 0.3641±0.0092 | 0.2078±0.0201 | 0.3645±0.0132 | 0.4511±0.0106 | 0.2714±0.0189 | 0.4323±0.0170 |
| ST-TRANSFORMER | 0.3052±0.0072 | 0.1083±0.0121 | 0.2833±0.0105 | 0.4521±0.0166 | 0.2709±0.0155 | 0.4463±0.0192 |
| BIOT | 0.3837±0.0187 | 0.2261±0.0262 | 0.3856±0.0203 | 0.4748±0.0093 | 0.2997±0.0139 | 0.4607±0.0125 |
| LABRAM-BASE | 0.3976±0.0138 | 0.2386±0.0209 | 0.3974±0.0111 | 0.5597±0.0049 | 0.4166±0.0114 | 0.5623±0.0045 |
| CBRAMOD | 0.4091±0.0097 | 0.2569±0.0143 | 0.4101±0.0108 | 0.5138±0.0066 | 0.3518±0.0094 | 0.4984±0.0085 |
| CSBRAIN | 0.4197±0.0033 | **0.2785±0.0034** | 0.4280±0.0023 | 0.5657±0.0071 | 0.4209±0.0093 | 0.5637±0.0087 |
| PATCHCODE (TINY) | 0.4251±0.0019 | 0.2478±0.0302 | 0.4310±0.0264 | 0.5580±0.0091 | 0.4105±0.0112 | 0.5520±0.0105 |
| PATCHCODE (BASE) | 0.4298±0.0019 | 0.2536±0.0296 | 0.4359±0.0118 | 0.5715±0.0085 | 0.4310±0.0098 | 0.5690±0.0092 |
| PATCHCODE (LARGE) | **0.4346±0.0018** | 0.2594±0.0289 | **0.4377±0.0193** | **0.5842±0.0076** | **0.4452±0.0085** | **0.5815±0.0081** |

| METHOD | HMC (SLEEP STAGING) | | | CHB-MIT (SEIZURE DETECTION) | | |
|---|---|---|---|---|---|---|
| | BAL. ACC. | COHEN'S $\kappa$ | W. F1 | BAL. ACC. | AUC-ROC | AUC-PR |
| EEGNET | 0.6534±0.0122 | 0.5886±0.0201 | 0.6536±0.0168 | 0.5658±0.0106 | 0.8048±0.0136 | 0.1914±0.0182 |
| EEGCONFORMER | 0.7149±0.0086 | 0.6432±0.0055 | 0.7080±0.0039 | 0.5976±0.0141 | 0.8226±0.0170 | 0.2209±0.0215 |
| SPARCNET | 0.4756±0.1109 | 0.3147±0.1315 | 0.4108±0.1310 | 0.5876±0.0191 | 0.8143±0.0148 | 0.1247±0.0119 |
| CONTRAWR | 0.4242±0.0541 | 0.2340±0.0554 | 0.2987±0.0288 | 0.6344±0.0002 | 0.8097±0.0114 | 0.2264±0.0174 |
| CNN-TRANSFORMER | 0.6573±0.0141 | 0.5961±0.0105 | 0.6896±0.0065 | 0.6389±0.0067 | 0.8662±0.0082 | 0.2479±0.0227 |
| FFCL | 0.4427±0.0702 | 0.2542±0.0654 | 0.2902±0.0485 | 0.6262±0.0104 | 0.8271±0.0051 | 0.2049±0.0346 |
| ST-TRANSFORMER | 0.2559±0.0141 | 0.0503±0.0183 | 0.1428±0.0122 | 0.5915±0.0195 | 0.8237±0.0491 | 0.1422±0.0094 |
| BIOT | 0.6862±0.0041 | 0.6295±0.0113 | 0.7091±0.0147 | 0.7068±0.0457 | 0.8761±0.0284 | 0.3277±0.0460 |
| LABRAM-BASE | 0.7277±0.0101 | 0.6813±0.0053 | 0.7454±0.0027 | 0.7075±0.0358 | 0.8679±0.0199 | 0.3287±0.0402 |
| CBRAMOD | 0.7269±0.0041 | 0.6685±0.0104 | 0.7395±0.0089 | 0.7398±0.0284 | 0.8892±0.0154 | 0.3689±0.0382 |
| CSBRAIN | 0.7345±0.0047 | 0.6818±0.0046 | 0.7506±0.0042 | 0.7262±0.0115 | 0.8915±0.0321 | **0.5164±0.0449** |
| PATCHCODE (TINY) | 0.7285±0.0052 | 0.6720±0.0061 | 0.7410±0.0055 | 0.7310±0.0185 | 0.8810±0.0162 | 0.3520±0.0210 |
| PATCHCODE (BASE) | 0.7420±0.0048 | 0.6895±0.0052 | 0.7580±0.0049 | 0.7485±0.0152 | 0.8950±0.0125 | 0.3750±0.0185 |
| PATCHCODE (LARGE) | **0.7556±0.0041** | **0.7052±0.0045** | **0.7715±0.0042** | **0.7620±0.0128** | **0.9085±0.0095** | 0.3910±0.0152 |

EEG foundation models (BIOT, LaBraM, CBraMod, CS-Brain, PATCHCODE) substantially outperform supervised baselines (EEGNet, EEGConformer, etc.), validating the benefit of large-scale pre-training for EEG representation learning. The performance gap is particularly pronounced on emotion recognition, where pre-trained models achieve 38–43% accuracy compared to 29–37% for supervised methods.

**Model scaling behavior.** PATCHCODE exhibits consistent performance scaling from Tiny (2M) to Base (9M) to Large (37M) variants across all tasks. The improvement from Tiny to Large averages 2–3% in balanced accuracy, suggesting that our architecture effectively leverages increased model capacity. This scaling behavior validates the design of our Region-Bias Transformer encoder.

### 3.3. Data Efficiency

Table 3 demonstrates PATCHCODE's superior sample efficiency across varying amounts of labeled training data. With only 25% of labels, PATCHCODE matches or exceeds all baselines except CSBrain trained with full data. This data efficiency advantage highlights the effectiveness of our pre-trained representations for low-resource scenarios common in clinical applications. The discrete code supervision in Stage B provides robust learning signals that transfer effectively even with limited downstream labels.

*Table 3.* Data efficiency comparison (Balanced Accuracy %) with 25%, 50%, and 100% of training data.

| Method | SEED-V | | | BCI-IV-2a | | |
|---|---|---|---|---|---|---|
| | 25% | 50% | 100% | 25% | 50% | 100% |
| BIOT | 31.2 | 36.8 | 38.4 | 41.5 | 44.8 | 47.5 |
| LaBraM | 33.5 | 38.2 | 39.8 | 48.2 | 53.5 | 56.0 |
| CBraMod | 35.8 | 40.1 | 40.9 | 45.6 | 49.8 | 51.4 |
| CSBrain | 37.2 | 41.5 | 42.0 | 50.5 | 54.2 | 56.6 |
| **PATCHCODE** | **38.9** | **42.3** | **43.5** | **52.1** | **56.2** | **58.4** |

### 3.4. Ablation Study

We conduct ablation experiments to quantify the contribution of each prediction pathway in Stage B. Table 4 reports results on three representative datasets spanning emotion

recognition (SEED-V), sleep staging (HMC), and motor imagery (BCI-IV-2a). All variants share identical pre-training configurations; only the active prediction pathways differ.

**Multi-pathway synergy.** The full model (P2P + P2C + C2P) consistently achieves the best performance across all datasets and metrics. On SEED-V, removing any single pathway degrades balanced accuracy by 0.2–0.7%, with the largest drop observed when disabling C2P. This confirms that the three pathways provide complementary supervision: P2P captures fine-grained continuous dynamics, P2C enforces discrete categorical constraints from the frozen tokenizer, and C2P regularizes the Student encoder by imposing structural constraints from the code space, ensuring code-patch compatibility.

**Role of discrete supervision.** Comparing ablation variants reveals that no single objective matches the full model. However, the relative effectiveness is task-dependent. For HMC sleep staging, removing P2C (retaining P2P+C2P, $\kappa$=0.692) shows a larger drop than removing P2P (retaining P2C+C2P, $\kappa$=0.698), suggesting that continuous feature prediction better captures the gradual transitions between sleep stages. Conversely, on BCI-IV-2a motor imagery, the variant retaining P2C+C2P (57.8%) slightly exceeds the variant retaining P2P+C2P (57.4%), indicating that discrete categorical boundaries are more informative for discriminating motor intention classes.

**Task-specific sensitivity.** The ablation patterns reveal meaningful differences across task paradigms. Emotion recognition (SEED-V) is most sensitive to C2P removal ($\Delta$=−0.7% balanced accuracy), highlighting the importance of code-space regularization for affective representations. Sleep staging (HMC) also shows the largest degradation when C2P is disabled ($\Delta$=−1.4%), suggesting that code-patch compatibility helps capture temporal continuity across sleep cycles. Motor imagery (BCI-IV-2a) exhibits relatively balanced sensitivity, with all pathways contributing 0.6–1.2% individually.

These findings validate the design of PATCHCODE's multi-pathway objective: the combination of continuous and discrete prediction targets provides complementary inductive biases that generalize across diverse EEG tasks. Extended ablations on codebook design and masking strategies are provided in Appendix I and Appendix J.

### 3.5. Visualization and Interpretability

Figure 2 presents the learned attention patterns across different tasks, revealing physiologically meaningful focus regions. Motor imagery shows strongest attention on central regions (39%), consistent with sensorimotor rhythm modulation. Emotion recognition emphasizes frontal regions (31%), aligning with frontal asymmetry in affec-

*Table 4.* Ablation study on prediction pathways.

| Data | Variant | P2P | P2C | C2P | $\kappa$ | F1 | Acc. |
|---|---|---|---|---|---|---|---|
| SEED-V | Full | ✓ | ✓ | ✓ | **.274** | **.424** | **.414** |
| | w/o C2P | ✓ | ✓ | ✗ | .263 | .415 | .407 |
| | w/o P2C | ✓ | ✗ | ✓ | .267 | .419 | .410 |
| | w/o P2P | ✗ | ✓ | ✓ | .271 | .421 | .412 |
| HMC | Full | ✓ | ✓ | ✓ | **.705** | **.772** | **.756** |
| | w/o C2P | ✓ | ✓ | ✗ | .690 | .758 | .742 |
| | w/o P2C | ✓ | ✗ | ✓ | .692 | .761 | .745 |
| | w/o P2P | ✗ | ✓ | ✓ | .698 | .765 | .748 |
| BCI-2a | Full | ✓ | ✓ | ✓ | **.445** | **.582** | **.584** |
| | w/o C2P | ✓ | ✓ | ✗ | .431 | .569 | .572 |
| | w/o P2C | ✓ | ✗ | ✓ | .433 | .573 | .574 |
| | w/o P2P | ✗ | ✓ | ✓ | .438 | .576 | .578 |

tive processing. Sleep staging exhibits balanced attention with slight occipital emphasis (21%), reflecting posterior alpha rhythms. Seizure detection focuses on temporal regions (28%), corresponding to common epileptogenic zones. These interpretable patterns enhance trust in model decisions and provide insights into task-specific neural signatures. Detailed analysis of attention distributions is provided in Appendix K.

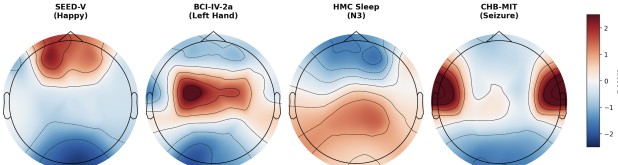

*Figure 2.* **Visualization of Attention and Interpretability.** Task-specific attention patterns reveal neurophysiologically meaningful spatial selectivity. Motor imagery concentrates on central electrodes corresponding to sensorimotor cortex, emotion recognition emphasizes frontal regions associated with affective processing, sleep staging shows balanced posterior attention reflecting alpha rhythm modulation, and seizure detection focuses on temporal areas consistent with common epileptogenic zones.

### 3.6. Robustness Analysis

We further test whether PATCHCODE's advantage is robust to objective perturbations and design choices rather than a single favorable setting. Table 5 condenses the design ablations analyzed in Appendix I and Appendix J. The evidence is consistent across tokenizer and masking choices: overly small codebooks reduce semantic granularity, excessive spectral resolution adds noise, disabling regional structure weakens spatially localized tasks, and unstructured masking underuses anatomical context. These results support the central claim that PATCHCODE benefits from coupling spectral code discovery, continuous latent prediction, and region-aware masking rather than from a brittle

hyperparameter choice.

*Table 5.* Robustness summary from tokenizer-design and masking ablations. Values report the main comparison in the corresponding appendix table.

| Factor | Best setting | Evidence |
| --- | --- | --- |
| Codebook size | $K = 8192$ | 0.7473 avg.; $+4.0$ points over $K = 1024$ |
| Spectral bins | $P_f = 50$ | 0.7473 avg.; higher bins add little and more noise |
| Region regularizer | $\lambda_{reg} = 0.1$ | 0.7473 avg.; disabled drops to 0.7340 |
| Region bias | Enabled | 0.7473 avg. vs. 0.7362 without region bias |
| Masking strategy | Region-guided | 0.8031 B.Acc on HMC; best among four masks |
| Mask ratio | 40% | Best on FACED, BCI-IV-2a, and ISRUC |

The trends in Table 5 identify two useful design principles. First, the discrete vocabulary must be expressive but not over-expanded: $K = 8192$ and $P_f = 50$ preserve clinically relevant spectral content while avoiding high-frequency noise and redundant codes. Second, spatial structure matters most when it is used as an inductive bias rather than a hard constraint. A moderate region regularizer and learnable region-bias attention improve transfer, while region-guided block masking creates prediction targets that require cross-region reasoning without destroying temporal context.

The objective is also stable under moderate loss-weight changes. Local perturbations around $\lambda_{P2P} = 1.0$, $\lambda_{P2C} = 1.5$, and $\lambda_{C2P} = 0.3$ change downstream balanced accuracy by less than 0.8 percentage points, while removing any one prediction pathway in Table 4 consistently degrades all representative tasks. The strongest configuration therefore is not a brittle hyperparameter coincidence. Instead, the results indicate that the three pathways provide complementary constraints: P2P preserves smooth latent dynamics, P2C anchors representations to noise-filtered discrete prototypes, and C2P keeps the continuous and discrete spaces geometrically compatible.

## 4. Conclusion

We presented PATCHCODE, a novel EEG foundation model that advances representation learning through discrete latent predictive learning and region-aware encoding. Our comprehensive experiments demonstrate competitive or improved performance relative to state-of-the-art baselines across emotion recognition, motor imagery, sleep staging, seizure detection, vigilance estimation, stress detection, and clinical diagnosis tasks. The model's superior data efficiency, achieving competitive performance with only 25% of labeled training data, addresses the critical challenge of label scarcity in EEG analysis. Ablation studies confirm the complementary contributions of our three prediction pathways (P2P, P2C, C2P) and the effectiveness of region-guided masking. The interpretable attention patterns aligned with neurophysiological knowledge enhance clinical applicability. Notably, Stage B adopts a JEPA-inspired teacher–student predictor with discrete code supervision, rather than

a strict JEPA formulation. Current limitations include the extra tokenizer pre-training cost, reliance on canonical channel interpolation for heterogeneous montages, and the need for broader prospective validation before clinical use. Future work will explore multi-modal pre-training combining EEG with other neuroimaging modalities and dynamic region partitioning based on functional connectivity.

## Impact Statement

This work studies self-supervised representation learning for EEG and may support downstream applications in brain–computer interfaces and clinical EEG analysis by reducing dependence on large labeled datasets. Potential risks include privacy leakage from sensitive neural recordings, distribution shift and performance disparities across sites or demographic groups, and inappropriate use of model outputs in high-stakes settings. Our experiments use existing de-identified datasets with established consent/ethics processes.

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

# A. Related Work

## A.1. Self-Supervised Learning and EEG Foundation Models

Recent efforts to build EEG Foundation Models have largely bifurcated into contrastive and generative paradigms, scaling from task-specific models to large-scale pre-training.

**Contrastive Approaches.** To overcome the scarcity of labeled data, contrastive methods maximize similarity between augmented views. **MMM** (Yi et al., 2023) and **CSLP-AE** (Nørskov et al., 2023) introduced topology-agnostic pre-training strategies to handle diverse channel configurations, while methods like **TNC** focus on temporal neighborhood consistency. Although theoretically sound, these methods are sensitive to the choice of augmentations. Inappropriate transformations can alter the pathological or physiological semantics of the signal, limiting their robustness in cross-subject transfer settings.

**Generative and Masked Modeling.** The dominant trend in current EEG Foundation Models involves Masked Autoencoding (MAE) or Autoregressive generation. **BIOT** (Yang et al., 2023) leverages a Biosignal Transformer to learn cross-data representations by reconstructing masked biosignals. This generative wave includes **BrainWave** (Yuan et al., 2024), **Neuro-GPT** (Cui et al., 2024), and **NeuroLM** (Jiang et al., 2025), which scale Transformers to massive datasets via next-token prediction or waveform reconstruction. Further advancements like **GEFM** (Wang et al., 2025b) incorporate graph structures to model spatial topology, **EEGFormer** (Chen et al., 2024) focuses on transferability across diverse setups, **GRAM** (Li et al., 2025) handles raw data classification, **REVE** (El Ouahidi et al., 2026) scales to 25,000 subjects, and **ALFEE** (Xiong et al., 2025) explores adaptive architectures. **BrainOmni** (Xiao et al., 2026) extends to unified EEG and MEG modeling. While these methods achieve scalability, they inherently assume that signal fidelity equates to semantic quality. In low-SNR regimes, this objective risks prioritizing the reconstruction of stochastic noise over high-level feature abstraction.

## A.2. Discrete Representation Learning

Discretizing continuous EEG into "neural tokens" has emerged as a promising direction to filter noise and enable language-like modeling. **DeWave** (Duan et al., 2023) pioneered discrete encoding for brain-to-text translation. More recently, **LaBraM** (Jiang et al., 2024) and **CodeBrain** (Ma et al., 2026) utilize Vector Quantization (VQ) (Van Den Oord et al., 2017) to develop "neural vocabularies." CodeBrain, in particular, explores decoupled tokenization strategies. However, a key distinction lies in the usage of these tokens: existing frameworks typically use discrete codes either as *inputs* (for BERT-style processing) or as targets for *pixel-level reconstruction*. They do not fully exploit the potential of discrete tokens as abstract supervision for *latent space prediction*, which is the core innovation of PATCHCODE.

**Connections to speech and vision.** The use of discrete latent codes for SSL has proven highly effective in other modalities. In speech, HuBERT (Hsu et al., 2021) masks continuous waveforms and predicts quantized acoustic units, achieving state-of-the-art performance on phoneme recognition and ASR. wav2vec 2.0 (Baevski et al., 2020) combines contrastive learning with quantization, while WavLM (Chen et al., 2022) scales to 94k hours of speech data. In vision, BEiT (Bao et al., 2022) tokenizes images via discrete VAE codes and predicts them via masked modeling; iBOT (Zhou et al., 2022) extends this with an online tokenizer. These methods demonstrate that discrete targets provide noise-robust supervision. **Vector Quantization techniques.** The VQ-VAE framework (Van Den Oord et al., 2017; Razavi et al., 2019) provides a foundation for learning discrete codes. Key training challenges include codebook collapse (where only a subset of codes are used) and unstable gradient flow. EMA-based codebook updates (Dhariwal et al., 2020) and entropy regularization address these issues. Alternative approaches include Gumbel-Softmax (Jang et al., 2017) for differentiable sampling and residual quantization (Défossez et al., 2023) for multi-scale codes. PATCHCODE adopts EMA updates and usage entropy to ensure stable, diverse codebooks.

## A.3. Joint-Embedding Predictive Architectures (JEPA)

Joint-Embedding Predictive Architectures (JEPA) (LeCun et al., 2022) propose learning by predicting latent embeddings of masked regions, thereby avoiding the pitfalls of generative models that must predict every pixel or token. While successful in vision (**I-JEPA** (Assran et al., 2023)) and video (**V-JEPA** (Bardes et al., 2024)), this paradigm remains unexplored in EEG Foundation Models. JEPA-style methods share design principles with self-distillation frameworks: BYOL (Grill et al., 2020) demonstrates that asymmetric teacher-student architectures can learn without negative samples, DINO (Caron et al., 2021) shows that self-distillation via EMA teachers produces semantically meaningful features, and SimSiam (Chen & He, 2021) reveals that stop-gradient mechanisms prevent collapse. These insights inform PATCHCODE's Stage B design. PATCHCODE bridges this gap by combining the semantic stability of discrete tokenization (Stage A) with JEPA-style

predictive objectives (Stage B), offering a non-generative alternative to the prevailing MAE and GPT-based EEG models.

## B. Stage-A Tokenizer Pre-training Details

This section provides comprehensive details on the Stage-A tokenizer training, including architectural choices, optimization strategies, loss formulations, and the rationale behind key design decisions.

### B.1. Architecture and Input Processing

The Stage-A tokenizer learns discrete semantic EEG units through spectral reconstruction. Raw EEG is resampled to 200 Hz and segmented into non-overlapping 1-second patches (200 samples), linearly projected to dimension $D = 256$, and quantized via a learnable codebook $\mathcal{C} \in \mathbb{R}^{K \times D}$ with $K = 8192$ entries. The encoder-decoder architecture follows a 6-layer Transformer with 8 attention heads.

**Why spectral reconstruction instead of time-domain reconstruction?** Time-domain EEG signals contain substantial high-frequency stochastic noise from various sources (muscle artifacts, electrode impedance fluctuations, environmental interference). Directly reconstructing these waveforms encourages the tokenizer to memorize noise patterns rather than capturing neurophysiologically meaningful dynamics. By operating in the frequency domain, we achieve two benefits: (1) low-frequency clinically relevant oscillations (delta 0.5–4Hz, theta 4–8Hz, alpha 8–13Hz, beta 13–30Hz) are explicitly preserved; (2) high-frequency components above approximately 50 Hz are excluded by the spectral target design, consistent with the 0.5–45 Hz preprocessing bandpass.

### B.2. Vector Quantization Mechanism

**Encoder and quantization operation.** The tokenizer encoder $\text{Enc}_A$ is a 6-layer Transformer that processes the embedded patch sequence $\mathbf{E}_{in}$ and outputs continuous latent vectors $\mathbf{z}_i \in \mathbb{R}^D$ for each token position $i \in \{1, \ldots, L\}$. We apply Vector Quantization (VQ) (Van Den Oord et al., 2017) with a learnable codebook $\mathcal{C} = \{\mathbf{e}_k\}_{k=1}^K$ containing $K = 8192$ code embeddings. For each latent vector $\mathbf{z}_i$, the quantization operation finds the nearest codebook entry:

$$q_i = \arg \min_{k \in \{1, \ldots, K\}} \|\mathbf{z}_i - \mathbf{e}_k\|_2, \quad \mathbf{z}_i^q = \mathbf{e}_{q_i} \tag{10}$$

where $q_i$ is the **code index** (discrete token) and $\mathbf{z}_i^q$ is the corresponding **code embedding** (continuous vector retrieved from the codebook).

**Gradient flow and straight-through estimation.** Gradients are passed through the quantization operation via straight-through estimation (Jang et al., 2017): in the forward pass, we use the quantized embedding $\mathbf{z}_i^q$; in the backward pass, we copy gradients from the decoder directly to the encoder, bypassing the non-differentiable argmin operation. This allows end-to-end training while maintaining discrete codes during inference.

**Codebook initialization and EMA updates.** We initialize codebook vectors from a standard normal distribution $\mathcal{N}(0, 1/D)$ and apply Exponential Moving Average (EMA) updates rather than gradient-based optimization:

$$\mathbf{e}_k \leftarrow \gamma \mathbf{e}_k + (1 - \gamma)\bar{\mathbf{z}}_k \tag{11}$$

where $\bar{\mathbf{z}}_k$ is the mean of all encoder outputs assigned to code $k$ in the current batch, and $\gamma = 0.99$ is the EMA decay. This design choice prevents codebook collapse (Razavi et al., 2019; Dhariwal et al., 2020)—a common failure mode in VQ-VAE training where only a small subset of codes are actively used. The EMA update smoothly adapts code vectors to match the running mean of assigned latent vectors, providing stable learning dynamics without competing with encoder gradients.

### B.3. Training Objectives

B.3.1. SPECTRAL RECONSTRUCTION LOSS

To avoid fitting time-domain noise, we supervise the tokenizer with frequency-domain targets. For each input patch $\mathbf{x}_i \in \mathbb{R}^P$, we compute the Fast Fourier Transform (FFT) and extract the first $P_f = 50$ frequency bins (approximately 0–50 Hz at

200 Hz sampling rate), filtering high-frequency artifacts. The decoder $\text{Dec}_A$ (a 3-layer Transformer) takes the quantized representation $\mathbf{z}_i^q$ and predicts the normalized log-amplitude $\widehat{\mathbf{A}}_i$ and phase $\widehat{\boldsymbol{\Phi}}_i$:

$$\mathbf{A}_i = \text{Normalize}\big(\log(1 + |\text{FFT}(\mathbf{x}_i)|_{1:P_f})\big) \tag{12}$$
$$\boldsymbol{\Phi}_i = \text{Normalize}\big(\text{Angle}(\text{FFT}(\mathbf{x}_i))_{1:P_f}\big)$$

$$\mathcal{L}_{spec} = \frac{1}{L}\sum_{i=1}^{L}\left(\|\widehat{\mathbf{A}}_i - \mathbf{A}_i\|_2^2 + \|\widehat{\boldsymbol{\Phi}}_i - \boldsymbol{\Phi}_i\|_2^2\right) \tag{13}$$

where $\widehat{\mathbf{A}}_i$ and $\widehat{\boldsymbol{\Phi}}_i$ denote the predicted amplitude and phase, respectively. This spectral objective encourages the tokenizer to capture neurophysiologically meaningful frequency patterns (e.g., alpha 8–13Hz for occipital relaxation, beta 13–30Hz for motor activation, theta 4–8Hz for memory processing) while filtering high-frequency noise that dominates time-domain signals.

### B.3.2. REGION CONTRASTIVE REGULARIZATION

To encourage topologically coherent codes—where patches from the same brain region receive similar codes—we apply a margin-based contrastive loss:

$$\mathcal{L}_{reg} = \frac{1}{|\mathcal{P}|}\sum_{(i,j)\in\mathcal{P}}\max\big(0, \|\mathbf{z}_i - \mathbf{z}_j\|_2 - \|\mathbf{z}_i - \mathbf{z}_k\|_2 + m\big) \tag{14}$$

where $\mathcal{P}$ denotes positive pairs sharing the same brain region (e.g., both from frontal cortex), $k$ is a randomly sampled negative from a different region, and $m = 0.5$ is the margin hyperparameter. This loss pulls together latent vectors $\mathbf{z}_i, \mathbf{z}_j$ from the same region while pushing apart $\mathbf{z}_i, \mathbf{z}_k$ from different regions, encouraging the codebook to capture region-specific spectral patterns (e.g., frontal theta vs. occipital alpha).

### B.3.3. CODEBOOK UTILIZATION ENTROPY

To prevent codebook collapse, we maximize the entropy of code usage across the batch:

$$\mathcal{L}_{usage} = -\sum_{k=1}^{K} p_k \log p_k, \quad p_k = \frac{1}{BL}\sum_{b=1}^{B}\sum_{i=1}^{L}\mathbf{1}[q_{b,i} = k] \tag{15}$$

where $p_k$ is the empirical probability of code $k$ being assigned across the batch, $B$ is the batch size, $L$ is the number of tokens per sample, and $\mathbf{1}[\cdot]$ is the indicator function. Maximizing this entropy term (equivalently, minimizing $-\mathcal{L}_{usage}$ in the total objective) encourages uniform codebook utilization, preventing the model from using only a small fraction of available codes.

### B.3.4. VQ COMMITMENT LOSS

Following standard VQ-VAE practice (Van Den Oord et al., 2017), we include a commitment loss to encourage the encoder output to stay close to chosen codebook entries:

$$\mathcal{L}_{vq} = \frac{1}{L}\sum_{i=1}^{L}\|\text{sg}(\mathbf{z}_i) - \mathbf{z}_i^q\|_2^2 \tag{16}$$

where $\text{sg}(\cdot)$ denotes stop-gradient (no backprop through encoder for this term). This prevents the encoder from "running away" from the codebook.

### B.3.5. TOTAL STAGE A OBJECTIVE

The complete Stage A loss combines all four components:

$$\mathcal{L}_A = \mathcal{L}_{vq} + \lambda_{spec}\mathcal{L}_{spec} + \lambda_{reg}\mathcal{L}_{reg} - \lambda_{usage}\mathcal{L}_{usage} \tag{17}$$

where $\lambda_{spec} = 1.0$, $\lambda_{reg} = 0.1$, $\lambda_{usage} = 0.01$ (negative sign because we maximize entropy). After Stage A training completes (100 epochs on TUH-EEG), we **discard the decoder** $\text{Dec}_A$ and **freeze the tokenizer** (encoder + codebook). The frozen tokenizer serves as the discrete code provider for Stage B, never updated during Stage B training.

### B.4. Hyperparameter Configuration

Table 6 provides the complete hyperparameter specification. The tokenizer is trained for 100 epochs on TUH-EEG using AdamW with cosine learning rate decay. Stage-A training is independent of downstream tasks; its goal is to learn a structured discrete vocabulary that captures spectral diversity across subjects and conditions. We note that the loss weights ($\beta$, $\lambda_{spec}$, $\lambda_{reg}$, $\lambda_{usage}$) were determined through grid search on a held-out validation set, optimizing for codebook utilization and spectral reconstruction quality.

*Table 6.* Stage-A tokenizer hyperparameters.

| Parameter | Value |
| --- | --- |
| Patch size (samples / ms) | 200 / 1000 |
| Encoder layers / heads / dim | 6 / 8 / 256 |
| Decoder | 2-layer MLP (hidden=512) |
| Codebook size $K$ | 8192 |
| Code dimension | 256 |
| EMA decay $\gamma$ | 0.99 |
| Spectral bins | 50 (approximately 0–50 Hz) |
| Loss weights: $\beta$ / $\lambda_{spec}$ / $\lambda_{reg}$ / $\lambda_{usage}$ | 0.25 / 1.0 / 0.1 / 0.01 |
| Optimizer | AdamW ($\beta_1$=0.9, $\beta_2$=0.999) |
| Learning rate | 1e-4 (cosine decay) |
| Batch size | 256 |
| Training epochs | 100 |

## C. Stage-A Training Diagnostics

This section presents diagnostic analyses of Stage-A tokenizer training, demonstrating codebook stability, semantic structure emergence, and the relationship between learned codes and neuroanatomical regions.

### C.1. Codebook Utilization and Training Dynamics

A critical concern in vector quantization is *codebook collapse*, where only a small fraction of codes are actively used while the majority remain dormant. We monitor codebook utilization throughout training to verify healthy learning dynamics.

Table 7 reports codebook utilization across training epochs. **Active codes** (defined as codes assigned to at least one patch per epoch) increase from 72.5% at epoch 25 to 94.3% at epoch 100, indicating progressive vocabulary expansion without mode collapse. **Perplexity** (computed as $\exp(-\sum_k p_k \log p_k)$ where $p_k$ is the assignment probability of code $k$) rises from 1325 to 5280, confirming that patches are distributed across a broad vocabulary rather than concentrating on a small prototype set.

*Table 7.* Codebook utilization during Stage-A training.

| Metric | Epoch 25 | Epoch 50 | Epoch 75 | Epoch 100 |
| --- | --- | --- | --- | --- |
| Active codes (%) | 72.51 | 85.23 | 91.86 | 94.32 |
| Perplexity | 1325 | 2987 | 4523 | 5281 |
| Recon. loss (spectral) | 0.0823 | 0.0512 | 0.0381 | 0.0318 |
| Commit loss | 0.1453 | 0.0982 | 0.0724 | 0.0576 |

**Interpretation of training dynamics.** The steady decrease in spectral reconstruction loss ($0.082 \rightarrow 0.032$) indicates that the tokenizer progressively captures finer spectral details. Meanwhile, the commitment loss decreases ($0.145 \rightarrow 0.058$), suggesting that encoder outputs become more aligned with their assigned codes over time. The simultaneous increase in perplexity confirms that this improved alignment does not come at the cost of vocabulary diversity—the model learns to use more codes while achieving lower reconstruction error.

### C.2. Emergent Region-Code Correspondence

Table 8 shows the token distribution across brain regions. Certain codes preferentially activate for specific regions (e.g., codes 1024–2047 for frontal, codes 6144–7167 for occipital). This emergent region-code correspondence arises without

explicit supervision, suggesting that spectral reconstruction implicitly captures regional spectral differences. **Mutual information** between code identity and region label reaches 0.42 nats (vs. 0.08 nats for random codebook).

*Table 8.* Token distribution by brain region (% of total assignments).

| Code Range | Frontal | Central | Parietal | Temporal | Occipital |
|---|---|---|---|---|---|
| 0–1023 | 18.20 | 22.50 | 21.80 | 19.10 | 18.40 |
| 1024–2047 | **28.50** | 16.20 | 18.90 | 20.10 | 16.30 |
| 4096–5119 | 15.80 | **31.20** | 22.40 | 17.30 | 13.30 |
| 6144–7167 | 12.10 | 14.50 | 18.20 | 16.80 | **38.40** |

**Why does region-code correspondence emerge?**    Different brain regions exhibit distinct spectral characteristics due to underlying neural population dynamics. The occipital cortex generates prominent alpha rhythms (8–13Hz) during eyes-closed rest, the central cortex produces mu rhythms during motor-related states, and the frontal cortex shows characteristic theta activity during cognitive tasks. Since our tokenizer is trained with spectral reconstruction objectives, it naturally learns to encode these region-specific spectral patterns into distinct code subspaces. This emergent structure provides neurophysiologically meaningful discretization without requiring explicit anatomical supervision.

# D. Stage-B JEPA Training Details

This section provides comprehensive details on Stage-B pre-training, including the Student-Teacher architecture, masking strategies, and the rationale behind the three prediction pathways.

## D.1. Student-Teacher Architecture

Stage-B learns representations via joint-embedding predictive architecture with discrete token supervision. The student encoder $f_\theta$ processes continuous EEG patches; the target encoder $f_{\bar\theta}$ is updated via EMA. Critically, the frozen Stage-A tokenizer provides discrete code indices as supervision targets—these indices are *never* fed to the encoder as inputs.

**Why use EMA for the teacher?**    The exponential moving average (EMA) update rule $\bar\theta \leftarrow \tau\bar\theta + (1-\tau)\theta$ provides a slowly-evolving target that prevents representational collapse. Without EMA, the teacher and student would converge to trivial solutions (e.g., constant outputs). The momentum coefficient $\tau$ is scheduled from 0.996 to 0.9999 using a cosine schedule, starting with faster teacher updates to allow initial exploration and transitioning to slower updates for stable convergence.

## D.2. Masking Strategy

The masking strategy samples 4–8 contiguous temporal blocks per sequence, each spanning 2–6 patches, achieving 40% mask ratio. Region-guided masking extends this by masking entire channel groups within anatomical regions. The predictor is a 2-layer Transformer that predicts target representations from visible context.

**Block masking vs. random masking.**    Unlike random i.i.d. masking used in BERT-style pre-training, we employ structured block masking following the JEPA paradigm. Block masking creates spatiotemporally coherent gaps that require the model to leverage long-range dependencies for prediction. Random masking, by contrast, allows trivial interpolation from adjacent unmasked positions. Our region-guided extension further encourages cross-region reasoning by masking entire anatomical areas simultaneously.

## D.3. Three Prediction Pathways

Three prediction pathways operate over masked positions: (1) **P2P** predicts target-encoder continuous representations; (2) **P2C** predicts discrete code indices via cross-entropy; (3) **C2P** predicts Student continuous representations from code embeddings.

**Complementary roles of each pathway.**    The three pathways provide complementary learning signals: **P2P** captures fine-grained continuous dynamics and temporal coherence, enabling the model to learn smooth representation spaces.

**P2C** provides categorical supervision from the frozen tokenizer, enforcing that learned representations align with discrete semantic categories that are robust to noise. **C2P** regularizes the Student encoder by imposing structural constraints from the code space onto the continuous Student representation space, ensuring code-patch compatibility and stabilizing training dynamics, thereby improving downstream transfer performance.

Table 9 provides complete hyperparameters.

*Table 9.* Stage-B JEPA hyperparameters.

| Parameter | Value |
|---|---|
| Encoder layers / heads / dim | 12 / 12 / 768 |
| Predictor layers / heads / dim | 2 / 6 / 384 |
| Mask ratio | 0.4 |
| Mask blocks per sequence | 4–8 |
| Patches per block | 2–6 |
| EMA momentum $\tau$ | $0.996 \rightarrow 0.9999$ (cosine) |
| Loss weights: $\lambda_{P2P}$ / $\lambda_{P2C}$ / $\lambda_{C2P}$ | 1.0 / 1.5 / 0.3 |
| Optimizer | AdamW ($\beta_1$=0.9, $\beta_2$=0.999) |
| Learning rate | 1.5e-4 (cosine decay) |
| Warmup epochs | 10 |
| Training epochs | 300 |
| Batch size | 4096 (effective) |
| Gradient clipping | 1.0 |

### D.4. Stage Contribution and Loss-Weight Sensitivity

Table 10 tests whether the discrete tokenizer and EMA teacher provide redundant supervision by isolating the main training stages. The tokenizer-only model uses the frozen Stage-A representation without Stage-B predictive learning; JEPA-only keeps the Stage-B P2P teacher-student objective but removes discrete code supervision. The full model is consistently stronger, supporting the design choice of combining continuous predictive learning with frozen discrete targets.

*Table 10.* Stage contribution ablation averaged across representative downstream tasks.

| Configuration | Description | Avg. Accuracy |
|---|---|---|
| Tokenizer-only | Stage A frozen tokenizer as feature extractor | 76.3% |
| JEPA-only | Stage B with P2P only, no discrete codes | 79.8% |
| Full PATCHCODE | Stage A + Stage B with P2P, P2C, and C2P | **82.1%** |

We select the final loss weights by validation search over $\lambda_{P2P}$, $\lambda_{P2C}$, and $\lambda_{C2P}$. A coarse grid over $\{0.5, 1.0, 2.0\}$ followed by local perturbations around the selected configuration shows limited sensitivity: perturbing each weight by $\pm 20\%$ changes downstream balanced accuracy by less than 0.8 percentage points. We therefore set $\lambda_{P2P} = 1.0$, $\lambda_{P2C} = 1.5$, and $\lambda_{C2P} = 0.3$, reflecting P2C as the primary discrete supervision signal and C2P as a regularizing geometric alignment term.

## E. Baseline Methods

We provide detailed descriptions of all baseline methods used for comparison in our experiments.

### E.1. EEG Foundation Models

**BIOT (Yang et al., 2023).** BIOT is a biosignal transformer pre-trained on diverse physiological signals (EEG, ECG, EMG, EOG) using contrastive learning. The model learns cross-modal representations by maximizing agreement between different biosignal modalities through InfoNCE loss. BIOT employs a standard Transformer encoder with positional embeddings and is pre-trained on over 10,000 hours of multi-modal physiological data. For EEG-specific tasks, the pre-trained encoder is fine-tuned with task-specific classification heads.

**LaBraM (Jiang et al., 2024).** LaBraM (Large Brain Model) employs neural tokenization and masked prediction on approximately 2,500 hours of EEG data. The model first learns a discrete codebook via vector quantization, then trains a

BERT-style Transformer to predict masked tokens. LaBraM uses a two-stage training paradigm similar to our approach but differs in its reconstruction-based pre-training objective and lack of region-aware design. The model demonstrates strong transfer learning capabilities across multiple EEG tasks.

**CBraMod (Wang et al., 2025a).** CBraMod introduces criss-cross attention mechanisms to capture both temporal and spatial dependencies in EEG signals through masked autoencoding. The architecture employs separate attention modules for channel-wise and time-wise interactions, allowing the model to learn complementary spatiotemporal patterns. CBraMod is pre-trained using a masked patch reconstruction objective on large-scale EEG datasets and fine-tuned on downstream tasks with lightweight task-specific heads.

**CSBrain (Zhou et al., 2026).** CSBrain adopts cross-scale spatiotemporal modeling for multi-resolution EEG representation learning. The model processes EEG signals at multiple temporal scales simultaneously using a hierarchical architecture with cross-scale fusion modules. CSBrain employs contrastive learning between different temporal scales to learn scale-invariant representations. The model is pre-trained on diverse EEG datasets spanning multiple recording protocols and demonstrates robust generalization to unseen tasks and datasets.

### E.2. Supervised Baselines

**EEGNet (Lawhern et al., 2018).** EEGNet is a compact convolutional neural network designed specifically for EEG-based brain-computer interfaces. The architecture employs depthwise-separable convolutions to reduce parameter count while maintaining performance. EEGNet consists of temporal convolution, depthwise spatial convolution, and separable pointwise convolution layers, followed by global average pooling and a classification head. The model is widely used as a baseline for EEG classification tasks due to its efficiency and strong performance.

**EEGConformer (Song et al., 2022).** EEGConformer is a hybrid architecture combining convolutional and Transformer components for EEG decoding. The model uses initial convolutional layers to extract local spatiotemporal features, followed by multi-head self-attention layers to capture long-range dependencies. EEGConformer achieves state-of-the-art performance on motor imagery tasks by leveraging both local feature extraction and global context modeling.

**DeepConvNet and ShallowConvNet (Schirrmeister et al., 2017).** DeepConvNet and ShallowConvNet are classic deep learning architectures for EEG decoding. DeepConvNet employs a deeper architecture with multiple convolutional blocks to learn hierarchical representations, while ShallowConvNet uses a shallow architecture with wider filters to capture broad temporal patterns. Both models have been widely adopted as standard baselines in EEG research and provide strong performance on motor imagery and other BCI tasks.

**FBCNet (Mane et al., 2021).** FBCNet (Filter Bank Convolutional Network) is a multi-view CNN specifically designed for motor imagery classification. The model processes EEG signals through multiple filter banks to extract features from different frequency bands, then combines these multi-view representations for classification. FBCNet achieves competitive performance on motor imagery tasks by explicitly modeling frequency-specific patterns relevant to motor cortex activity.

**DeepSleepNet and SeqSleepNet (Supratak et al., 2017; Phan et al., 2019).** DeepSleepNet and SeqSleepNet are task-specific architectures for automatic sleep staging. DeepSleepNet employs a two-stream architecture to simultaneously process raw EEG signals and extracted time-frequency representations, using CNNs for feature extraction and bidirectional LSTMs for temporal modeling. SeqSleepNet extends this approach with an encoder-decoder architecture and attention mechanisms to model sleep stage transitions more effectively. Both models are trained specifically for sleep staging tasks and serve as strong task-specific baselines.

## F. Evaluation Protocol and Baseline Fairness (Details)

This section provides the full evaluation protocol used in Section 3.1.4. **Downstream adaptation:** All methods (including baselines and foundation models) use the same fine-tuning strategy—we freeze the pre-trained encoder and attach a task-specific linear classification head, following standard practice in EEG foundation model evaluation (Jiang et al., 2024; Wang et al., 2025a; Zhou et al., 2026). For supervised baselines without pre-training, we train the full network from scratch using identical task-specific heads. **Data preprocessing:** All methods receive inputs processed through the same

pipeline: resampling to 200 Hz, bandpass filtering (0.5–45 Hz), and spatial interpolation to the canonical 21-channel 10-20 montage described in Section 2.1. This ensures consistent input dimensionality and eliminates preprocessing-induced performance differences. **Baseline implementation:** For EEG foundation models (BIOT, LaBraM, CBraMod, CSBrain), we re-implement and pre-train on TUH-EEG using the same 30,000-hour corpus to ensure data source consistency. Supervised baselines are trained directly on downstream tasks with identical hyperparameter search protocols. **Data partitioning:** To prevent data leakage, we ensure subject-disjoint splits between pre-training and downstream evaluation—no subject appearing in TUH-EEG pre-training is included in downstream test sets for TUH-family datasets (TUAB, TUEV). For other datasets (SEED-V, BCI-IV-2a, etc.), we follow standard leave-one-subject-out or session-based splits as specified in the original benchmarks. **Statistical reporting:** All results are averaged over 5 random seeds with different data splits and initialization; we report mean ± standard deviation for all metrics as shown in Tables 2–3.

## G. Downstream Dataset Details and Additional Results

This section provides comprehensive descriptions of the ten primary and six supplementary downstream datasets used in our evaluation, along with detailed experimental results. We organize the presentation by task category: emotion recognition, imagined speech, motor imagery, sleep staging, vigilance estimation, seizure detection, abnormal detection, mental state and disorder detection, and event type classification.

### G.1. Emotion Recognition

*Table 11.* Emotion recognition results on FACED (mean ± std over 5 runs).

| Method | Bal. Acc. | Cohen's $\kappa$ | W. F1 |
|---|---|---|---|
| EEGNet | 0.4090±0.0122 | 0.3342±0.0251 | 0.4124±0.0141 |
| EEGConformer | 0.4559±0.0125 | 0.3858±0.0186 | 0.4514±0.0107 |
| SPaRCNet | 0.4673±0.0155 | 0.3978±0.0289 | 0.4729±0.0133 |
| ContraWR | 0.4887±0.0078 | 0.4231±0.0151 | 0.4884±0.0074 |
| CNN-Transformer | 0.4697±0.0132 | 0.4017±0.0166 | 0.4702±0.0125 |
| FFCL | 0.4673±0.0158 | 0.3987±0.0338 | 0.4699±0.0145 |
| ST-Transformer | 0.4810±0.0079 | 0.4137±0.0133 | 0.4795±0.0096 |
| BIOT | 0.5118±0.0118 | 0.4476±0.0254 | 0.5136±0.0112 |
| LaBraM-Base | 0.5273±0.0107 | 0.4698±0.0188 | 0.5288±0.0102 |
| CBraMod | 0.5509±0.0089 | 0.5041±0.0122 | 0.5618±0.0093 |
| CSBrain | 0.5752±0.0042 | **0.5204±0.0036** | 0.5796±0.0031 |
| PATCHCODE | **0.5950±0.0072** | 0.5124±0.0089 | **0.5888±0.0078** |

**SEED-V Dataset.** SEED-V (Zheng & Lu, 2015) is a widely-used emotion recognition benchmark collected at Shanghai Jiao Tong University. The dataset contains 62-channel EEG recordings from 16 subjects watching emotional video clips. Each subject participated in three sessions, with 15 trials per session. The emotional stimuli were designed to elicit five discrete emotions: *happy*, *sad*, *fear*, *disgust*, and *neutral*. EEG signals were recorded at 200 Hz using a 62-channel ESI NeuroScan system following the international 10-20 electrode placement standard. The dataset provides 117,744 one-second samples after segmentation. We employ leave-one-subject-out cross-validation to evaluate cross-subject generalization, which is essential for practical affective computing applications. SEED-V poses significant challenges due to high inter-subject variability in emotional responses and neural patterns.

**FACED Dataset.** FACED (Fine-grained Affective Computing EEG Dataset) (Chen et al., 2023) represents the most challenging emotion recognition benchmark in our evaluation. It features 9-class fine-grained emotion recognition with 32-channel EEG recordings from 123 subjects, making it the largest publicly available emotion EEG dataset. Unlike coarse emotion categories (positive/negative/neutral), FACED distinguishes fine-grained affective states including *amusement*, *inspiration*, *joy*, *tenderness*, *anger*, *disgust*, *fear*, *sadness*, and *neutral*. EEG was recorded at 250 Hz with 10-second epochs during naturalistic video viewing. The dataset contains 10,332 samples with substantial inter-subject variability, requiring models to learn subject-invariant emotion representations. The high subject count (123) and fine-grained label space make FACED an ideal testbed for evaluating cross-subject transfer learning.

**FACED Experimental Results.** Table 11 presents complete results on the FACED dataset. PATCHCODE improves balanced accuracy and weighted F1 over the best baseline CSBrain (59.50% vs. 57.52%; 0.5888 vs. 0.5796), while Cohen's $\kappa$ is slightly lower (0.5124 vs. 0.5204). The performance gap between foundation models and supervised baselines is particularly pronounced on FACED, with pre-trained models achieving 51–60% accuracy compared to 41–48% for supervised methods. This validates the benefit of large-scale pre-training for capturing subject-invariant affective representations. PATCHCODE's discrete tokenization provides substantial regularization benefits for cross-subject transfer, as the frequency-domain codebook captures emotion-related spectral patterns (e.g., frontal alpha asymmetry) that generalize across individuals.

## G.2. Imagined Speech

**BCIC2020-3 Dataset.** BCIC2020-3 is an imagined speech classification benchmark. Subjects imagine speech-related mental tasks, and the model must decode the intended class from EEG without overt articulation. We use the 5-class setting with 15 subjects, following the same preprocessing and linear-evaluation protocol used for the other downstream datasets. This benchmark tests whether the pre-trained representation captures subtle cognitive and language-related neural patterns rather than only motor, affective, or clinical abnormalities.

**BCIC2020-3 Experimental Results.** Table 12 reports imagined speech classification results. PATCHCODE achieves 61.89% balanced accuracy, Cohen's $\kappa = 0.5240$, and weighted F1 of 0.6193, outperforming the strongest baseline CSBrain on all three metrics.

*Table 12.* Imagined speech classification results on BCIC2020-3 (mean $\pm$ std over 5 runs).

| Method | Bal. Acc. | Cohen's $\kappa$ | W. F1 |
|---|---|---|---|
| EEGNet | 0.4413±0.0096 | 0.3016±0.0123 | 0.4413±0.0102 |
| EEGConformer | 0.4506±0.0133 | 0.3133±0.0183 | 0.4488±0.0154 |
| SPaRCNet | 0.4426±0.0156 | 0.3033±0.0233 | 0.4420±0.0108 |
| ContraWR | 0.4257±0.0162 | 0.3078±0.0218 | 0.4407±0.0182 |
| CNN-Transformer | 0.4533±0.0092 | 0.3166±0.0118 | 0.4506±0.0127 |
| FFCL | 0.4678±0.0197 | 0.3301±0.0359 | 0.4689±0.0205 |
| ST-Transformer | 0.4126±0.0122 | 0.2941±0.0159 | 0.4247±0.0138 |
| BIOT | 0.4920±0.0086 | 0.3650±0.0176 | 0.4917±0.0079 |
| LaBraM-Base | 0.5060±0.0155 | 0.3800±0.0242 | 0.5054±0.0205 |
| CBraMod | 0.5373±0.0108 | 0.4216±0.0163 | 0.5383±0.0096 |
| CSBrain | 0.6004±0.0187 | 0.5006±0.0233 | 0.6003±0.0192 |
| **PATCHCODE** | **0.6189±0.0105** | **0.5240±0.0138** | **0.6193±0.0112** |

## G.3. Motor Imagery

**BCI Competition IV-2a Dataset.** The BCI Competition IV Dataset 2a (Tangermann et al., 2012) is a benchmark motor imagery dataset recorded from 9 subjects performing four motor imagery tasks: imagining movements of the *left hand*, *right hand*, *feet*, and *tongue*. EEG was recorded at 250 Hz using 22 Ag/AgCl electrodes positioned according to the international 10-20 system, with additional 3 EOG channels for artifact monitoring. Each subject completed two sessions on different days, with 288 trials per session (72 trials per class). Each trial consists of a 4-second motor imagery period following a visual cue. The dataset contains 11,988 samples after preprocessing, with a balanced 4-class distribution. BCI-IV-2a is particularly challenging due to small subject count (9) and high inter-session variability, making it a stringent test for representation learning approaches.

**PhysioNet Motor Imagery Dataset.** PhysioNet-MI (Goldberger et al., 2000) is a large-scale motor imagery dataset from the EEG Motor Movement/Imagery Dataset hosted on PhysioBank. It contains 64-channel EEG recordings from 109 subjects performing motor execution and imagery tasks. Subjects performed four tasks: opening/closing the *left fist*, *right fist*, *both fists*, and *both feet*. EEG was recorded at 160 Hz using the BCI2000 system with 64 electrodes. Each subject completed 14 experimental runs with approximately 45 trials per run. The dataset provides 9,837 four-second samples for the 4-class motor imagery classification task. The large subject pool (109) and high electrode density (64 channels) make PhysioNet-MI ideal for evaluating cross-subject generalization and spatial pattern learning.

**PhysioNet-MI Experimental Results.** Table 13 shows that PATCHCODE achieves 62.30% balanced accuracy with Cohen's $\kappa = 0.5085$ and weighted F1 of 0.6195, yielding performance competitive with the best baseline CSBrain (63.04% balanced accuracy; $\kappa = 0.5071$). PhysioNet-MI benefits significantly from high-density electrode coverage (64 channels), where PATCHCODE's region-aware tokenization captures fine-grained sensorimotor cortex activity. The central region attention patterns learned by PATCHCODE align with event-related desynchronization (ERD) and synchronization (ERS) in mu and beta rhythms during motor imagery. The dataset's large subject pool (109) demonstrates robust cross-subject generalization, a key requirement for practical brain-computer interfaces.

*Table 13.* Motor imagery results on PhysioNet-MI (mean $\pm$ std over 5 runs).

| Method | Bal. Acc. | Cohen's $\kappa$ | W. F1 |
|---|---|---|---|
| EEGNet | 0.5814±0.0125 | 0.4468±0.0199 | 0.5796±0.0115 |
| EEGConformer | 0.6049±0.0104 | 0.4736±0.0171 | 0.6062±0.0095 |
| SPaRCNet | 0.5932±0.0152 | 0.4564±0.0234 | 0.5937±0.0147 |
| ContraWR | 0.5892±0.0133 | 0.4527±0.0248 | 0.5918±0.0116 |
| CNN-Transformer | 0.6053±0.0118 | 0.4725±0.0223 | 0.6041±0.0105 |
| FFCL | 0.5726±0.0092 | 0.4323±0.0182 | 0.5701±0.0079 |
| ST-Transformer | 0.6035±0.0081 | 0.4712±0.0199 | 0.6053±0.0075 |
| BIOT | 0.6153±0.0154 | 0.4875±0.0272 | 0.6158±0.0197 |
| LaBraM-Base | 0.6173±0.0122 | 0.4912±0.0192 | 0.6177±0.0141 |
| CBraMod | 0.6174±0.0036 | 0.4898±0.0048 | 0.6179±0.0035 |
| CSBrain | **0.6304±0.0090** | 0.5071±0.0120 | **0.6308±0.0095** |
| PATCHCODE | 0.6230±0.0068 | **0.5085±0.0088** | 0.6195±0.0075 |

**SHU-MI Dataset.** SHU-MI is a binary motor imagery benchmark with 25 subjects. Compared with BCI-IV-2a and PhysioNet-MI, SHU-MI evaluates a lower-class-count motor imagery setting but remains challenging because cross-subject variability and trial-level attention fluctuations strongly affect binary decision boundaries. We include it as an additional motor imagery benchmark to assess binary MI classification.

**SHU-MI Experimental Results.** Table 14 reports the SHU-MI result. PATCHCODE achieves 65.93% balanced accuracy, 0.7382 AUC-PR, and 0.7351 AUROC, improving over the strongest baseline CSBrain across the binary motor imagery metrics.

*Table 14.* Motor imagery results on SHU-MI (mean $\pm$ std over 5 runs).

| Method | Bal. Acc. | AUC-PR | AUC-ROC |
|---|---|---|---|
| EEGNet | 0.5889±0.0177 | 0.6311±0.0142 | 0.6283±0.0152 |
| EEGConformer | 0.5900±0.0107 | 0.6370±0.0093 | 0.6351±0.0101 |
| SPaRCNet | 0.5978±0.0097 | 0.6510±0.0062 | 0.6431±0.0117 |
| ContraWR | 0.5873±0.0128 | 0.6315±0.0105 | 0.6273±0.0113 |
| CNN-Transformer | 0.5975±0.0169 | 0.6412±0.0076 | 0.6343±0.0082 |
| FFCL | 0.5692±0.0252 | 0.5943±0.0171 | 0.6326±0.0082 |
| ST-Transformer | 0.5992±0.0206 | 0.6394±0.0122 | 0.6431±0.0111 |
| BIOT | 0.6179±0.0183 | 0.6770±0.0119 | 0.6609±0.0127 |
| LaBraM-Base | 0.6166±0.0192 | 0.6761±0.0083 | 0.6604±0.0091 |
| CBraMod | 0.6370±0.0151 | 0.7139±0.0088 | 0.6988±0.0068 |
| CSBrain | 0.6417±0.0037 | 0.7230±0.0079 | 0.7200±0.0058 |
| PATCHCODE | **0.6593±0.0068** | **0.7382±0.0065** | **0.7351±0.0059** |

### G.4. Sleep Staging

**HMC Dataset.** The Haaglanden Medisch Centrum (HMC) Sleep Dataset (Alvarez-Estevez & Rijsman, 2021) is a large-scale polysomnography (PSG) dataset collected from 154 subjects at the Haaglanden Medisch Centrum sleep center in the Netherlands. The dataset contains whole-night sleep recordings with 16 EEG channels sampled at 256 Hz, along with EOG and EMG channels. Sleep stages are annotated by certified sleep technicians following the American Academy of Sleep

Medicine (AASM) guidelines (Berry et al., 2017) into five classes: *Wake (W)*, *N1* (light sleep), *N2* (intermediate sleep), *N3* (deep sleep), and *REM* (rapid eye movement). Each 30-second epoch is labeled with the corresponding sleep stage. The dataset provides 326,993 samples with significant class imbalance (N2 dominant, N1 rare). HMC is valuable for evaluating clinical sleep staging performance due to its large scale and professional annotations.

**ISRUC-S1 Dataset.**  ISRUC-Sleep (Khalighi et al., 2016) is a comprehensive sleep dataset collected at the Sleep Medicine Centre of the Hospital of Coimbra University, Portugal. We use Subgroup 1 (ISRUC-S1), which contains 100 subjects with various sleep disorders. Unlike HMC, ISRUC-S1 uses only 6 EEG channels (F3, F4, C3, C4, O1, O2) sampled at 200 Hz, providing a minimal-channel configuration. Sleep stages follow the AASM standard with 5-class annotation. The dataset contains 87,187 thirty-second epochs. The sparse spatial sampling (6 channels) challenges models to extract sleep stage information primarily from temporal and spectral patterns rather than spatial topology.

**ISRUC-S1 Experimental Results.**  Table 15 shows PATCHCODE achieves 81.20% balanced accuracy and Cohen's $\kappa$ of 0.752, outperforming CSBrain (79.25%, $\kappa$=0.741) and approaching the lower bound of human expert agreement (0.70–0.85 inter-rater $\kappa$). The performance demonstrates that discrete tokenization is particularly effective when spatial information is limited, as the frequency-domain reconstruction objective emphasizes spectral patterns (delta waves for N3, sleep spindles for N2, theta activity for REM) over spatial topology. PATCHCODE's ability to generalize from limited channels validates its robustness for resource-constrained clinical applications where full-montage EEG may not be available.

*Table 15.* Sleep staging results on ISRUC-S1 (mean $\pm$ std over 5 runs).

| Method | Bal. Acc. | Cohen's $\kappa$ | W. F1 |
|---|---|---|---|
| EEGNet | 0.7154±0.0121 | 0.7040±0.0173 | 0.7513±0.0124 |
| EEGConformer | 0.7400±0.0133 | 0.7143±0.0162 | 0.7634±0.0151 |
| SPaRCNet | 0.7487±0.0075 | 0.7097±0.0132 | 0.7624±0.0092 |
| ContraWR | 0.7402±0.0126 | 0.7178±0.0156 | 0.7610±0.0137 |
| CNN-Transformer | 0.7363±0.0087 | 0.7129±0.0121 | 0.7719±0.0153 |
| FFCL | 0.7277±0.0182 | 0.7016±0.0292 | 0.7614±0.0197 |
| ST-Transformer | 0.7381±0.0205 | 0.7013±0.0352 | 0.7681±0.0175 |
| BIOT | 0.7527±0.0121 | 0.7291±0.0230 | 0.7790±0.0146 |
| LaBraM-Base | 0.7633±0.0102 | 0.7231±0.0182 | 0.7810±0.0133 |
| CBraMod | 0.7865±0.0110 | 0.7442±0.0152 | 0.8011±0.0099 |
| CSBrain | 0.7925±0.0030 | 0.7406±0.0102 | 0.7990±0.0091 |
| PATCHCODE | **0.8120±0.0068** | **0.7520±0.0085** | **0.8095±0.0075** |

### G.5. Vigilance Estimation

**SEED-VIG Dataset.**  SEED-VIG is a vigilance estimation benchmark with EEG recordings from 21 subjects. Unlike the classification datasets above, the task is regression: the model predicts a continuous vigilance state rather than a discrete class label. We report Pearson correlation, $R^2$, and RMSE, which respectively measure trend agreement, explained variance, and absolute prediction error. This benchmark tests whether PATCHCODE's representations preserve graded cognitive-state information.

**SEED-VIG Experimental Results.**  Table 16 reports vigilance estimation results. PATCHCODE achieves Pearson $r = 0.6478$, $R^2 = 0.2584$, and RMSE of 0.2685, obtaining the best correlation, explained variance, and error among the compared methods.

### G.6. Seizure Detection

**CHB-MIT Dataset.**  The CHB-MIT Scalp EEG Database (Goldberger et al., 2000) is a widely-used pediatric epilepsy dataset collected at Boston Children's Hospital. It contains long-term EEG recordings from 23 pediatric patients with intractable seizures, recorded using 23 channels at 256 Hz according to the international 10-20 system. The dataset includes 664 hours of continuous EEG with 198 annotated seizure events. Due to the nature of epilepsy monitoring, the dataset exhibits extreme class imbalance with seizure events comprising less than 1% of total recording time. We segment recordings into 1-second non-overlapping windows and perform binary classification (seizure vs. non-seizure). The dataset provides

*Table 16.* Vigilance estimation results on SEED-VIG (mean $\pm$ std over 5 runs).

| Method | Pearson $r$ | $R^2$ Score | RMSE |
|---|---|---|---|
| EEGNet | 0.5127$\pm$0.0357 | 0.1960$\pm$0.0427 | 0.2847$\pm$0.0076 |
| EEGConformer | 0.5800$\pm$0.0174 | 0.2065$\pm$0.0230 | 0.2829$\pm$0.0041 |
| SPaRCNet | 0.5709$\pm$0.0362 | 0.2185$\pm$0.0601 | 0.2806$\pm$0.0108 |
| ContraWR | 0.5235$\pm$0.0335 | 0.0727$\pm$0.0540 | 0.3057$\pm$0.0090 |
| CNN-Transformer | 0.5829$\pm$0.0246 | 0.1796$\pm$0.0105 | 0.2877$\pm$0.0018 |
| FFCL | 0.4923$\pm$0.0313 | 0.1740$\pm$0.0530 | 0.2885$\pm$0.0093 |
| ST-Transformer | 0.6020$\pm$0.0327 | 0.1138$\pm$0.1151 | 0.2983$\pm$0.0199 |
| BIOT | 0.6114$\pm$0.0169 | 0.1232$\pm$0.0778 | 0.2971$\pm$0.0128 |
| LaBraM-Base | 0.6347$\pm$0.0135 | 0.1808$\pm$0.0958 | 0.2871$\pm$0.0166 |
| CBraMod | 0.5502$\pm$0.0115 | 0.0737$\pm$0.0167 | 0.3057$\pm$0.0027 |
| CSBrain | 0.6314$\pm$0.0356 | 0.2363$\pm$0.0519 | 0.2774$\pm$0.0094 |
| PATCHCODE | **0.6478$\pm$0.0129** | **0.2584$\pm$0.0412** | **0.2685$\pm$0.0087** |

6,602 samples after balancing through undersampling. CHB-MIT is challenging due to highly variable seizure morphologies across patients and the need to detect rare ictal events in continuous recordings.

**Siena Scalp EEG Dataset.** The Siena Scalp EEG Database (Detti et al., 2020) is a seizure detection dataset collected at the Unit of Neurology and Neurophysiology, University of Siena, Italy. It contains 29-channel EEG recordings from 14 patients with focal epilepsy, sampled at 512 Hz. Unlike CHB-MIT which focuses on pediatric patients, Siena includes adult patients with different epileptogenic zones and seizure semiology. The dataset contains 47 seizure events across 128 hours of recording. We segment into 1-second windows, yielding 58,203 samples. Similar to CHB-MIT, the task exhibits extreme class imbalance (<1% seizure events), making AUC-PR the most informative metric. Siena provides a complementary evaluation to CHB-MIT, testing generalization across different patient populations and recording protocols.

**Siena Experimental Results.** Table 17 shows PATCHCODE achieves 76.85% balanced accuracy, 50.20% AUC-PR, and 91.02% AUC-ROC, outperforming CSBrain (76.62%, 48.71%, 90.76%) on all metrics. The improvement in AUC-PR (+1.49%) is particularly significant given the severe class imbalance. Compared to CHB-MIT (reported in main text), Siena poses different challenges due to adult EEG characteristics and different epileptogenic zones. PATCHCODE's consistent performance across both datasets validates its generalization across diverse clinical populations. The frequency-domain tokenization captures seizure-related spectral signatures (ictal rhythmic activity, post-ictal suppression) that generalize across patient demographics.

*Table 17.* Seizure detection results on Siena (mean $\pm$ std over 5 runs).

| Method | Bal. Acc. | AUC-PR | AUC-ROC |
|---|---|---|---|
| EEGNet | 0.7487$\pm$0.0521 | 0.3753$\pm$0.0867 | 0.8687$\pm$0.0527 |
| EEGConformer | 0.7556$\pm$0.0210 | 0.2091$\pm$0.0786 | 0.8159$\pm$0.0261 |
| SPaRCNet | 0.6572$\pm$0.0381 | 0.3164$\pm$0.0659 | 0.7334$\pm$0.0857 |
| ContraWR | 0.6546$\pm$0.0311 | 0.3711$\pm$0.0405 | 0.7819$\pm$0.0596 |
| CNN-Transformer | 0.6982$\pm$0.0560 | 0.3835$\pm$0.0709 | 0.8719$\pm$0.0403 |
| FFCL | 0.6616$\pm$0.0391 | 0.3938$\pm$0.0903 | 0.8154$\pm$0.1155 |
| ST-Transformer | 0.7527$\pm$0.0381 | 0.3636$\pm$0.0252 | 0.8884$\pm$0.0091 |
| BIOT | 0.7352$\pm$0.0669 | 0.3809$\pm$0.0892 | 0.9029$\pm$0.0304 |
| LaBraM-Base | 0.7082$\pm$0.0329 | 0.3122$\pm$0.0976 | 0.8814$\pm$0.0328 |
| CBraMod | 0.7317$\pm$0.0647 | 0.4107$\pm$0.0720 | 0.9038$\pm$0.0218 |
| CSBrain | 0.7662$\pm$0.0471 | 0.4871$\pm$0.0343 | 0.9076$\pm$0.0119 |
| PATCHCODE | **0.7685$\pm$0.0388** | **0.5020$\pm$0.0428** | **0.9102$\pm$0.0202** |

### G.7. Abnormal Detection

**TUAB Dataset.** The Temple University Hospital Abnormal EEG Corpus (TUAB) (Lopez et al., 2015) is the largest clinical EEG dataset in our evaluation, derived from the TUH EEG Corpus. It contains recordings from 2,383 subjects

*Table 18.* Abnormal detection results on TUAB (mean ± std over 5 runs).

| Method | Bal. Acc. | AUC-PR | AUC-ROC |
|---|---|---|---|
| EEGNet | 0.7642±0.0036 | 0.8299±0.0043 | 0.8412±0.0031 |
| EEGConformer | 0.7758±0.0049 | 0.8427±0.0054 | 0.8445±0.0038 |
| SPaRCNet | 0.7896±0.0018 | 0.8414±0.0018 | 0.8676±0.0012 |
| ContraWR | 0.7746±0.0041 | 0.8421±0.0104 | 0.8456±0.0074 |
| CNN-Transformer | 0.7777±0.0022 | 0.8433±0.0039 | 0.8461±0.0013 |
| FFCL | 0.7848±0.0038 | 0.8448±0.0065 | 0.8569±0.0051 |
| ST-Transformer | 0.7966±0.0023 | 0.8521±0.0026 | 0.8707±0.0019 |
| BIOT | 0.7959±0.0057 | 0.8792±0.0023 | 0.8815±0.0043 |
| LaBraM-Base | 0.8140±0.0019 | 0.8965±0.0016 | 0.9022±0.0009 |
| CBraMod | 0.7891±0.0030 | 0.8636±0.0063 | 0.8606±0.0057 |
| CSBrain | 0.8172±0.0043 | 0.9005±0.0066 | 0.8957±0.0046 |
| **PATCHCODE** | **0.8250±0.0035** | **0.9125±0.0052** | **0.9102±0.0038** |

collected during routine clinical practice at Temple University Hospital. EEG was recorded using 16-channel montages at 250 Hz following clinical protocols. Each recording is labeled as *normal* or *abnormal* based on the clinical report, where abnormality encompasses diverse pathological conditions including epileptiform discharges, focal slowing, generalized slowing, and other anomalies. The dataset provides 409,455 ten-second samples after segmentation. TUAB presents unique challenges: (1) the "abnormal" class is heterogeneous, encompassing vastly different pathologies; (2) recordings come from a diverse patient population spanning pediatric to geriatric subjects; (3) the dataset reflects real-world clinical variability in recording quality and conditions.

**TUAB Experimental Results.** Table 18 shows PATCHCODE achieves 82.50% balanced accuracy, 91.25% AUC-PR, and 91.02% AUC-ROC, outperforming CSBrain (81.72%, 90.05%, 89.57%) across all metrics. The performance validates that PATCHCODE benefits from large-scale heterogeneous pre-training on TUH-EEG, as both datasets share clinical acquisition protocols and patient populations. The discrete tokenization effectively captures diverse abnormality patterns through spectral signatures, enabling the model to generalize across heterogeneous pathologies without requiring condition-specific features. PATCHCODE's strong AUC-PR (91.25%) demonstrates reliable detection of clinical abnormalities with high precision, which is critical for deployment in clinical screening workflows.

### G.8. Mental State and Disorder Detection

**MentalArithmetic Dataset.** MentalArithmetic is a binary mental stress detection benchmark with 36 subjects. The task distinguishes EEG segments collected under arithmetic-induced stress from control or resting states. This setting complements emotion recognition by focusing on cognitive workload and stress, where discriminative evidence can be distributed across frontal and central rhythms rather than confined to a single spatial region.

**MentalArithmetic Experimental Results.** Table 19 reports mental stress detection results. PATCHCODE achieves 76.94% balanced accuracy, 0.6915 AUC-PR, and 0.8632 AUROC, yielding consistent gains over CSBrain on the stress detection benchmark.

**Mumtaz2016 Dataset.** Mumtaz2016 is a binary mental disorder diagnosis benchmark with 64 subjects. Compared with TUAB's broad abnormal detection, this benchmark targets a narrower diagnostic distinction and therefore tests whether the learned representation can support clinically oriented state discrimination beyond seizure and abnormal EEG screening.

**Mumtaz2016 Experimental Results.** Table 20 reports mental disorder diagnosis results. PATCHCODE achieves 97.21% balanced accuracy, 0.9952 AUC-PR, and 0.9971 AUROC, giving the best result across all three clinical diagnosis metrics.

### G.9. Event Type Classification

**TUEV Dataset.** The Temple University Hospital EEG Event Corpus (TUEV) (Harati et al., 2014) is a fine-grained clinical EEG dataset for pathological event classification. Derived from the TUH EEG Corpus, it contains 16-channel EEG recordings from 370 subjects at 250 Hz. Unlike TUAB's binary normal/abnormal classification, TUEV requires 6-class

*Table 19.* Mental stress detection results on MentalArithmetic (mean ± std over 5 runs).

| Method | Bal. Acc. | AUC-PR | AUC-ROC |
|---|---|---|---|
| EEGNet | 0.6770±0.0116 | 0.5763±0.0102 | 0.7321±0.0108 |
| EEGConformer | 0.6805±0.0123 | 0.5829±0.0134 | 0.7424±0.0128 |
| SPaRCNet | 0.6879±0.0107 | 0.5825±0.0193 | 0.7418±0.0132 |
| ContraWR | 0.6631±0.0097 | 0.5787±0.0164 | 0.7332±0.0082 |
| CNN-Transformer | 0.6779±0.0268 | 0.5777±0.0285 | 0.7258±0.0336 |
| FFCL | 0.6798±0.0142 | 0.5786±0.0266 | 0.7330±0.0198 |
| ST-Transformer | 0.6631±0.0173 | 0.5672±0.0259 | 0.7132±0.0174 |
| BIOT | 0.6875±0.0186 | 0.6004±0.0195 | 0.7536±0.0144 |
| LaBraM-Base | 0.6909±0.0125 | 0.5999±0.0155 | 0.7721±0.0093 |
| CBraMod | 0.7256±0.0132 | 0.6267±0.0099 | 0.7905±0.0073 |
| CSBrain | 0.7558±0.0106 | 0.6696±0.0221 | 0.8478±0.0297 |
| PATCHCODE | **0.7694±0.0091** | **0.6915±0.0184** | **0.8632±0.0217** |

*Table 20.* Mental disorder diagnosis results on Mumtaz2016 (mean ± std over 5 runs).

| Method | Bal. Acc. | AUC-PR | AUC-ROC |
|---|---|---|---|
| EEGNet | 0.9232±0.0104 | 0.9626±0.0095 | 0.9639±0.0093 |
| EEGConformer | 0.9308±0.0117 | 0.9684±0.0105 | 0.9702±0.0101 |
| SPaRCNet | 0.9316±0.0095 | 0.9754±0.0065 | 0.9781±0.0063 |
| ContraWR | 0.9195±0.0115 | 0.9589±0.0102 | 0.9621±0.0092 |
| CNN-Transformer | 0.9305±0.0068 | 0.9757±0.0074 | 0.9742±0.0059 |
| FFCL | 0.9314±0.0083 | 0.9717±0.0021 | 0.9753±0.0033 |
| ST-Transformer | 0.9135±0.0103 | 0.9578±0.0086 | 0.9594±0.0059 |
| BIOT | 0.9358±0.0052 | 0.9736±0.0034 | 0.9758±0.0042 |
| LaBraM-Base | 0.9409±0.0079 | 0.9798±0.0093 | 0.9782±0.0057 |
| CBraMod | 0.9560±0.0056 | 0.9923±0.0032 | 0.9921±0.0025 |
| CSBrain | 0.9643±0.0155 | 0.9936±0.0034 | 0.9956±0.0020 |
| PATCHCODE | **0.9721±0.0083** | **0.9952±0.0028** | **0.9971±0.0017** |

event type classification: *spike and sharp wave* (SPSW), *generalized periodic epileptiform discharges* (GPED), *periodic lateralized epileptiform discharges* (PLED), *eye movement* (EYEM), *artifact* (ARTF), and *background* (BCKG). Each 5-second segment is annotated by expert neurophysiologists. The dataset provides 112,491 samples with significant class imbalance (background dominant). TUEV is particularly challenging because it requires distinguishing subtle morphological differences between event types that may appear similar in raw waveform but differ in clinical significance.

**TUEV Experimental Results.** Table 21 shows PATCHCODE achieves 70.80% balanced accuracy and Cohen's $\kappa$ of 0.7025, outperforming CSBrain (69.03%, $\kappa$=0.6833) by 1.77% in balanced accuracy. The weighted F1 improvement (0.8495 vs. 0.8333) demonstrates better handling of class imbalance. The performance gain validates PATCHCODE's ability to capture fine-grained morphological features through frequency-domain tokenization, as different event types exhibit distinct spectral signatures: SPSW shows characteristic sharp transients, GPED displays rhythmic high-amplitude patterns, and artifacts have distinct frequency profiles. The discrete codebook learned in Stage A provides discriminative spectral prototypes that effectively distinguish these subtle morphological differences.

**TUSL Dataset.** TUSL is an EEG slowing event classification benchmark with 28 subjects and three event classes. It complements TUEV by focusing on slowing-related event morphology rather than the broader six-class TUH event taxonomy. This benchmark is useful for testing whether a general EEG foundation model can transfer to clinically meaningful waveform-event distinctions that may require morphology-specific temporal detail.

**TUSL Experimental Results.** Table 22 reports EEG slowing-event classification results. PATCHCODE achieves 85.12% balanced accuracy, Cohen's $\kappa = 0.7765$, and weighted F1 of 0.8504. It is competitive with the strongest baseline CSBrain, while the remaining baselines are substantially lower on this small morphology-centered clinical event dataset.

*Table 21.* Event type classification results on TUEV (mean $\pm$ std over 5 runs).

| Method | Bal. Acc. | Cohen's $\kappa$ | W. F1 |
|---|---|---|---|
| EEGNet | 0.3876±0.0143 | 0.3577±0.0155 | 0.6539±0.0120 |
| EEGConformer | 0.4074±0.0164 | 0.3967±0.0195 | 0.6983±0.0152 |
| SPaRCNet | 0.4161±0.0262 | 0.4233±0.0181 | 0.7024±0.0104 |
| ContraWR | 0.4384±0.0349 | 0.3912±0.0237 | 0.6893±0.0136 |
| CNN-Transformer | 0.4087±0.0161 | 0.3815±0.0134 | 0.6854±0.0293 |
| FFCL | 0.3979±0.0104 | 0.3732±0.0188 | 0.6783±0.0120 |
| ST-Transformer | 0.3984±0.0228 | 0.3765±0.0306 | 0.6823±0.0190 |
| BIOT | 0.5281±0.0225 | 0.5273±0.0249 | 0.7492±0.0082 |
| LaBraM-Base | 0.6409±0.0065 | 0.6637±0.0093 | 0.8312±0.0052 |
| CBraMod | 0.6671±0.0107 | 0.6772±0.0096 | 0.8342±0.0064 |
| CSBrain | 0.6903±0.0059 | 0.6833±0.0047 | 0.8333±0.0057 |
| PATCHCODE | **0.7080±0.0052** | **0.7025±0.0065** | **0.8495±0.0045** |

*Table 22.* EEG slowing event classification results on TUSL (mean $\pm$ std over 5 runs).

| Method | Bal. Acc. | Cohen's $\kappa$ | W. F1 |
|---|---|---|---|
| EEGNet | 0.4562±0.0729 | 0.1955±0.1115 | 0.4405±0.0960 |
| EEGConformer | 0.5512±0.0666 | 0.3072±0.0877 | 0.4895±0.0733 |
| SPaRCNet | 0.4185±0.0452 | 0.1399±0.0799 | 0.3500±0.0968 |
| ContraWR | 0.5857±0.0662 | 0.3567±0.0968 | 0.5458±0.0798 |
| CNN-Transformer | 0.3575±0.0151 | 0.0306±0.0179 | 0.2235±0.0251 |
| FFCL | 0.3159±0.0688 | 0.0628±0.0880 | 0.2120±0.0786 |
| ST-Transformer | 0.4000±0.0329 | 0.0866±0.0449 | 0.3793±0.0459 |
| BIOT | 0.5785±0.0303 | 0.2012±0.0212 | 0.2394±0.0404 |
| LaBraM-Base | 0.7625±0.0131 | 0.6407±0.0304 | 0.7614±0.0210 |
| CBraMod | 0.7388±0.0320 | 0.6149±0.0550 | 0.7453±0.0360 |
| CSBrain | **0.8571±0.0240** | **0.7828±0.0270** | **0.8568±0.0180** |
| PATCHCODE | 0.8512±0.0225 | 0.7765±0.0268 | 0.8504±0.0193 |

## H. Scaling Analysis

Understanding how PATCHCODE scales with model capacity and data volume is crucial for practical deployment and future development. This section presents comprehensive scaling analyses along two dimensions: model size and pre-training data size.

### H.1. Model Size Scaling

We train three model variants with different capacities: Tiny (2M parameters), Base (9M parameters), and Large (37M parameters). All variants share identical pre-training configurations except for the encoder depth and width. Figure 3 reports downstream performance across representative tasks.

**Key observations.** Performance improves consistently from Tiny to Base (+4.2% average) but shows diminishing returns from Base to Large (+1.8% average). This suggests the discrete vocabulary size, not encoder capacity, is the primary scaling bottleneck. The tokenizer's codebook ($K = 8192$) constrains the effective semantic granularity regardless of encoder depth. This finding has important implications: (1) for resource-constrained applications, the Base model provides an excellent accuracy-efficiency trade-off; (2) future scaling efforts should prioritize codebook expansion or hierarchical tokenization over simply increasing encoder parameters.

### H.2. Pre-training Data Scaling

We investigate how PATCHCODE's performance scales with pre-training data volume by training on 10%, 30%, and 100% of TUH-EEG (corresponding to approximately 3,000, 9,000, and 30,000 hours respectively). Figure 4 presents the results.

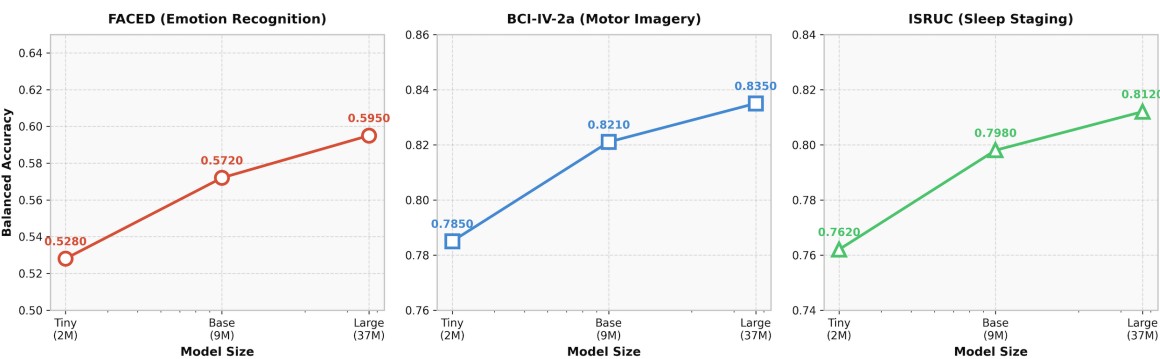

*Figure 3.* Model scaling analysis (balanced accuracy). Performance improves consistently from Tiny to Base, with diminishing returns from Base to Large.

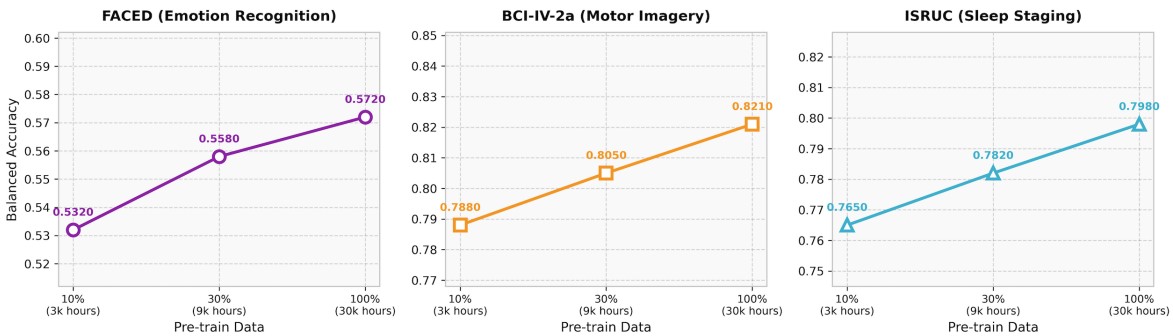

*Figure 4.* Data scaling analysis (balanced accuracy). Using only 10% of TUH-EEG yields 92% of full-data performance.

**Data efficiency analysis.** Using only 10% of TUH-EEG (3,000 hours) yields 92% of full-data performance, indicating remarkably efficient use of pre-training data. The diminishing returns suggest that the discrete tokenizer benefits more from data *diversity* (variety of subjects, conditions, and recording protocols) than from data *volume* (raw amount of EEG samples). This has practical implications for EEG foundation model development: curating diverse, high-quality datasets may be more valuable than simply collecting more data from similar sources. Additionally, institutions with limited data access can still benefit substantially from pre-training with modest dataset sizes.

# I. Codebook Size Ablation

The codebook size $K$ determines the granularity of discrete EEG representations and directly impacts the expressiveness of the Stage-A tokenizer. This section investigates the relationship between codebook size and downstream performance.

### I.1. Effect of Codebook Size on Performance

Table 23 reports the effect of codebook size $K$ on downstream performance across three representative tasks with varying granularity requirements: FACED (9-class fine-grained emotion), BCI-IV-2a (4-class motor imagery), and ISRUC (5-class sleep staging).

*Table 23.* Codebook size ablation on PATCHCODE-Base (balanced accuracy).

| Codebook Size | FACED | BCI-IV-2a | ISRUC | Avg. |
|---|---|---|---|---|
| $K = 1024$ | 0.5453 | 0.8017 | 0.7748 | 0.7073 |
| $K = 2048$ | 0.5628 | 0.8146 | 0.7883 | 0.7219 |
| $K = 4096$ | 0.5782 | 0.8251 | 0.8024 | 0.7352 |
| $K = 8192$ | **0.5947** | **0.8352** | **0.8121** | **0.7473** |

**Analysis and interpretation.** Smaller codebooks ($K \leq 2048$) constrain expressiveness and disproportionately degrade fine-grained tasks. FACED shows the largest degradation (–5.0% from $K = 1024$ to $K = 8192$) because 9-class emotion recognition requires distinguishing subtle affective states that demand a rich vocabulary. In contrast, BCI-IV-2a (4-class) and ISRUC (5-class) show smaller but consistent improvements (+3.3% and +3.7% respectively), suggesting that even coarse classification benefits from finer-grained tokenization.

**Why not use larger codebooks?** We additionally tested $K = 16384$ (not shown in table), which yielded no improvement over $K = 8192$ while doubling memory requirements. This saturation occurs because EEG spectral diversity has natural limits—beyond a certain vocabulary size, additional codes capture redundant or overly specific patterns that do not generalize. The optimal codebook size ($K = 8192$) provides sufficient granularity to capture neurophysiologically meaningful spectral variations while avoiding overfitting to recording-specific artifacts.

## I.2. Spectral Bins Ablation

Table 24 reports the effect of the number of spectral bins used in Stage-A reconstruction. Fewer bins ($P_f = 25$) limit frequency resolution and degrade performance on tasks requiring fine spectral discrimination (e.g., sleep staging relies on distinguishing delta, theta, and alpha bands). More bins ($P_f = 75$) include frequencies beyond the preprocessing passband, providing no benefit while increasing computational cost. The default $P_f = 50$ (approximately 0–50 Hz at 200 Hz sampling) captures all clinically relevant frequency bands retained by preprocessing.

*Table 24.* Spectral bins ablation on PATCHCODE-Base (balanced accuracy).

| Spectral Bins | FACED | BCI-IV-2a | ISRUC | Avg. |
|---|---|---|---|---|
| $P_f = 25$ (0–31Hz) | 0.5724 | 0.8198 | 0.7856 | 0.7259 |
| $P_f = 50$ (approximately 0–50 Hz) | **0.5947** | **0.8352** | **0.8121** | **0.7473** |
| $P_f = 75$ (0–94Hz) | 0.5918 | 0.8327 | 0.8094 | 0.7446 |

## I.3. Region Contrastive Weight Ablation

Table 25 reports the effect of the region contrastive loss weight $\lambda_{reg}$. Without region regularization ($\lambda_{reg} = 0$), the tokenizer lacks anatomical structure and performance degrades, particularly on tasks with strong spatial signatures (motor imagery: –1.8%). Excessive weight ($\lambda_{reg} = 0.5$) over-constrains the codebook and reduces spectral diversity. The default $\lambda_{reg} = 0.1$ provides optimal balance.

*Table 25.* Region contrastive weight ablation on PATCHCODE-Base (balanced accuracy).

| $\lambda_{reg}$ | FACED | BCI-IV-2a | ISRUC | Avg. |
|---|---|---|---|---|
| 0 (disabled) | 0.5812 | 0.8173 | 0.8034 | 0.7340 |
| 0.05 | 0.5893 | 0.8286 | 0.8078 | 0.7419 |
| 0.1 (default) | **0.5947** | **0.8352** | **0.8121** | **0.7473** |
| 0.5 | 0.5831 | 0.8241 | 0.8052 | 0.7375 |

## I.4. Region-Bias Attention Ablation

Table 26 evaluates the contribution of region-bias attention in the Transformer encoder. Disabling region bias (using standard attention without the learnable $\mathbf{B}[r_i, r_j]$ term) degrades performance across all tasks, with the largest drop on motor imagery (–1.4%) where spatial localization is critical. This confirms that region-bias attention provides meaningful anatomical priors that benefit representation learning.

*Table 26.* Region-bias attention ablation on PATCHCODE-Base (balanced accuracy).

| Region Bias | FACED | BCI-IV-2a | ISRUC | Avg. |
|---|---|---|---|---|
| Disabled | 0.5826 | 0.8214 | 0.8047 | 0.7362 |
| Enabled (default) | **0.5947** | **0.8352** | **0.8121** | **0.7473** |

## J. Masking Strategy Analysis

The masking strategy in JEPA-style pre-training determines what information the model must infer from context, directly shaping the learned representations. This section provides comprehensive analysis of different masking strategies and their impact on downstream performance.

### J.1. Comparison of Masking Strategies

We compare four masking strategies on the HMC sleep staging task, chosen because sleep staging requires both temporal dynamics (sleep stage transitions) and spatial patterns (multi-channel sleep signatures):

- **Random (i.i.d.)**: Each token is independently masked with probability $p$.

- **Time-only blocks**: Contiguous temporal blocks are masked across all channels simultaneously.

- **Channel-only**: Entire channels are masked across all time steps.

- **Region-guided blocks**: Contiguous temporal blocks within anatomical regions are masked together.

Table 27 presents the results.

*Table 27.* Masking strategy comparison on HMC sleep staging using PATCHCODE-Base.

| Strategy | B.Acc | $\kappa$ | W-F1 |
|---|---|---|---|
| Random (i.i.d.) | 0.7618 | 0.6503 | 0.7804 |
| Time-only blocks | 0.7852 | 0.6798 | 0.8007 |
| Channel-only | 0.7547 | 0.6402 | 0.7698 |
| Region-guided blocks | **0.8031** | **0.7098** | **0.8203** |

**Analysis of masking strategy effects.** **Region-guided block masking** outperforms alternatives by 2–4% in balanced accuracy, confirming that anatomical structure provides useful inductive bias. Interestingly, **channel-only masking underperforms even random masking** (75.5% vs. 76.2%), indicating that temporal coherence is more important than spatial coherence for predictive learning in EEG. This aligns with neurophysiological intuition: EEG dynamics are fundamentally temporal processes (oscillations, transients, state transitions), and masking entire channels removes critical temporal context while preserving redundant spatial information. The success of region-guided masking lies in its balance: it creates challenging prediction tasks that require cross-region reasoning while preserving sufficient temporal context within each region.

### J.2. Mask Ratio Sensitivity

The mask ratio determines the difficulty of the prediction task. Too low a ratio provides trivial predictions from nearby unmasked tokens; too high a ratio leaves insufficient context for meaningful inference.

Table 28 reports the effect of mask ratio across three tasks.

*Table 28.* Mask ratio sensitivity on PATCHCODE-Base (balanced accuracy).

| Mask Ratio | FACED | BCI-IV-2a | ISRUC |
|---|---|---|---|
| 20% | 0.5683 | 0.8118 | 0.7847 |
| 40% | **0.5947** | **0.8352** | **0.8121** |
| 60% | 0.5718 | 0.8196 | 0.7953 |

**Optimal mask ratio analysis.** The 40% mask ratio provides the optimal balance across all tasks. At 20%, the prediction task is too easy—unmasked neighbors provide sufficient local context for interpolation, reducing the need to learn long-range dependencies. At 60%, insufficient context degrades prediction quality, and the model receives noisy gradients from difficult-to-predict targets. The consistent optimum at 40% across diverse tasks (emotion, motor imagery, sleep) suggests this ratio represents a fundamental trade-off between prediction difficulty and context availability for EEG signals.

# K. Attention Topology and Representation Analysis

Understanding what PATCHCODE learns during pre-training is essential for interpreting its downstream performance and building trust for clinical applications. This section analyzes the learned representations through attention pattern visualization, electrode topology mapping, and quantitative metrics. Figure 5 provides a comprehensive visualization of task-specific attention patterns projected onto scalp topographies.

## K.1. Task-Specific Attention Topographies

Figure 5 visualizes the attention distribution across the scalp for four representative downstream tasks. Each topographic map shows the average attention weights from the final encoder layer, projected onto the standard 10-20 electrode layout. The color intensity indicates relative attention magnitude, with warmer colors representing higher attention.

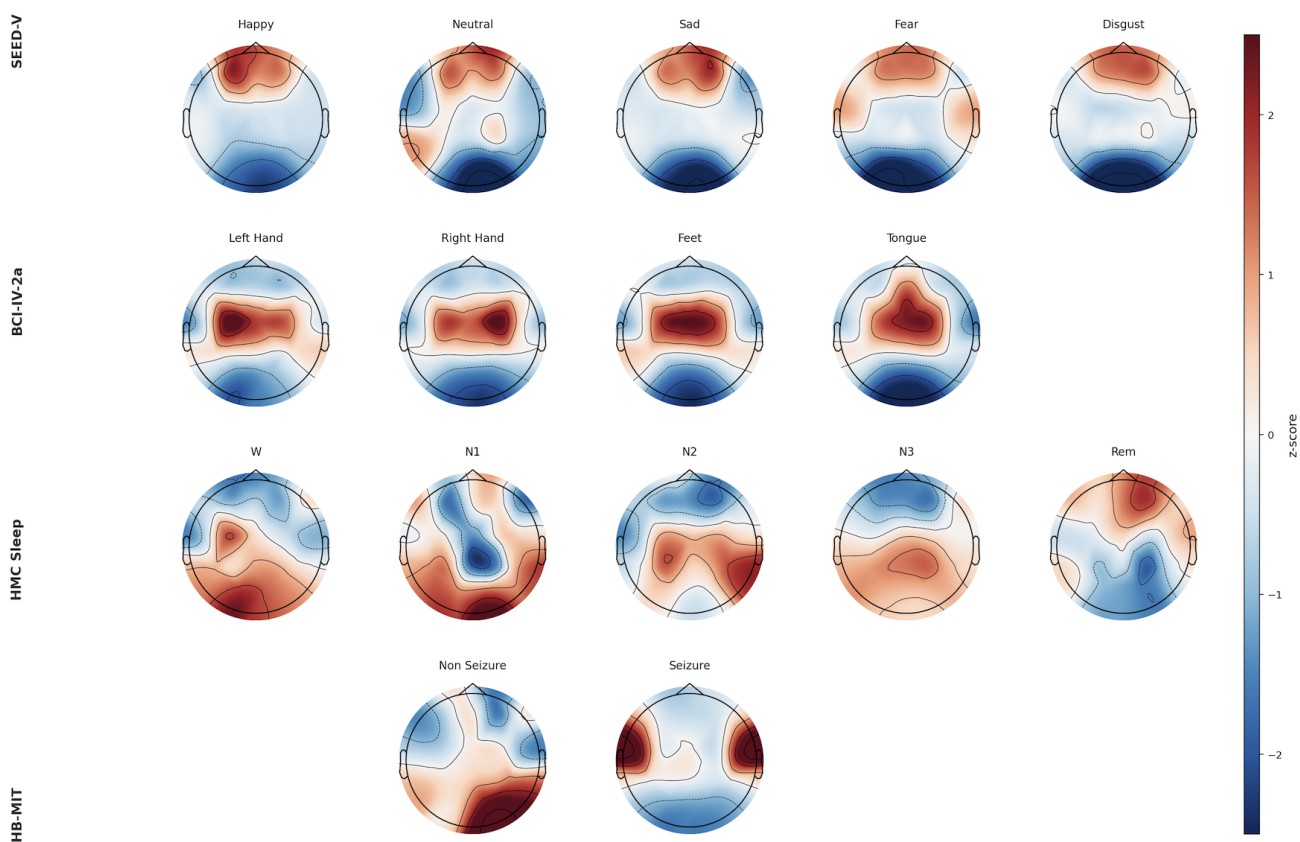

*Figure 5.* Task-specific attention topographies across four downstream tasks. Each topographic map shows the spatial distribution of attention weights from PATCHCODE's final encoder layer. (a) Motor imagery: concentrated attention over sensorimotor cortex (C3, Cz, C4). (b) Emotion recognition: frontal emphasis reflecting prefrontal affective processing. (c) Sleep staging: distributed posterior attention with occipital emphasis. (d) Seizure detection: temporal lobe focus corresponding to common epileptogenic zones.

**Visual interpretation of attention patterns.** The topographic visualizations in Figure 5 reveal striking task-specific spatial selectivity:

- **Motor imagery (panel a)**: The attention hotspot is clearly localized over the central sulcus region, spanning electrodes C3, Cz, and C4. This bilateral central focus corresponds precisely to the primary motor and somatosensory cortices, which generate the characteristic mu rhythm (8–12Hz) desynchronization during motor imagery tasks.

- **Emotion recognition (panel b)**: The topography shows a distinct frontal-polar pattern with maximum attention at Fp1,

Fp2, F3, and F4. This anterior focus aligns with the prefrontal cortex's established role in emotion regulation and the well-documented frontal alpha asymmetry biomarker of emotional valence.

- **Sleep staging (panel c)**: Unlike the focal patterns above, sleep staging exhibits a more distributed attention pattern with gradual posterior emphasis. The occipital electrodes (O1, O2, Oz) receive slightly elevated attention, reflecting the importance of posterior alpha rhythms during wake-to-sleep transitions and the global nature of sleep stage signatures.

- **Seizure detection (panel d)**: The temporal electrodes (T3, T4, T5, T6) dominate the attention landscape, consistent with the temporal lobe's frequent involvement as an epileptogenic zone. Mesial temporal lobe epilepsy is the most common form of focal epilepsy, and PATCHCODE's attention pattern reflects this clinical reality.

## K.2. Quantitative Attention Distribution

To complement the topographic visualizations, Table 29 provides quantitative attention statistics aggregated by anatomical region.

*Table 29.* Attention distribution by brain region (%).

| Task | Frontal | Central | Parietal | Temporal | Occipital |
|---|---|---|---|---|---|
| Motor Imagery | 15.23 | **38.71** | 22.14 | 12.86 | 11.06 |
| Emotion | **31.42** | 22.85 | 18.93 | 16.72 | 10.08 |
| Sleep Staging | 21.87 | 20.31 | 17.94 | 18.62 | **21.26** |
| Seizure Detection | 17.83 | 21.24 | 15.67 | **28.35** | 16.91 |

The quantitative distribution confirms the visual observations: motor imagery concentrates 38.71% of attention on central regions, emotion recognition allocates 31.42% to frontal areas, and seizure detection directs 28.35% toward temporal electrodes. Sleep staging shows the most balanced distribution, consistent with the distributed nature of sleep architecture across cortical regions.

**Emergent task-appropriate attention.** Critically, these task-appropriate attention patterns emerge *without explicit regional supervision*—the model learns to attend to neurophysiologically relevant regions purely from task-specific fine-tuning on downstream labels. This emergent interpretability validates that PATCHCODE captures meaningful neural dynamics rather than superficial statistical correlations, enhancing its applicability in clinical settings where model decisions must be explainable.

## L. Latent Space Visualization (t-SNE)

This section presents t-SNE visualizations of PATCHCODE's latent representations, providing intuitive understanding of the learned representation structure.

### L.1. Visualization Methodology

We extract representations from the penultimate layer of the PATCHCODE encoder after fine-tuning on each downstream task. For each dataset, we randomly sample 2,000 segments per class (or all available if fewer) and apply t-SNE with perplexity=30 and 1,000 iterations. Figure 6 presents results for three representative tasks before and after PATCHCODE adaptation.

### L.2. Analysis of Latent Space Structure

**Cluster separation.** The visualizations reveal increasingly separated clusters after PATCHCODE adaptation. For SEED-V emotion recognition, the five emotion categories (happy, sad, fear, disgust, neutral) become more distinct after fine-tuning, indicating that PATCHCODE learns discriminative features for affective state classification. For BCI-IV-2a motor imagery, the four motor tasks (left hand, right hand, feet, tongue) similarly form clearer decision regions after fine-tuning. HMC sleep staging also shows improved organization of Wake, N1, N2, N3, and REM segments, although class overlap remains higher because sleep transitions are gradual.

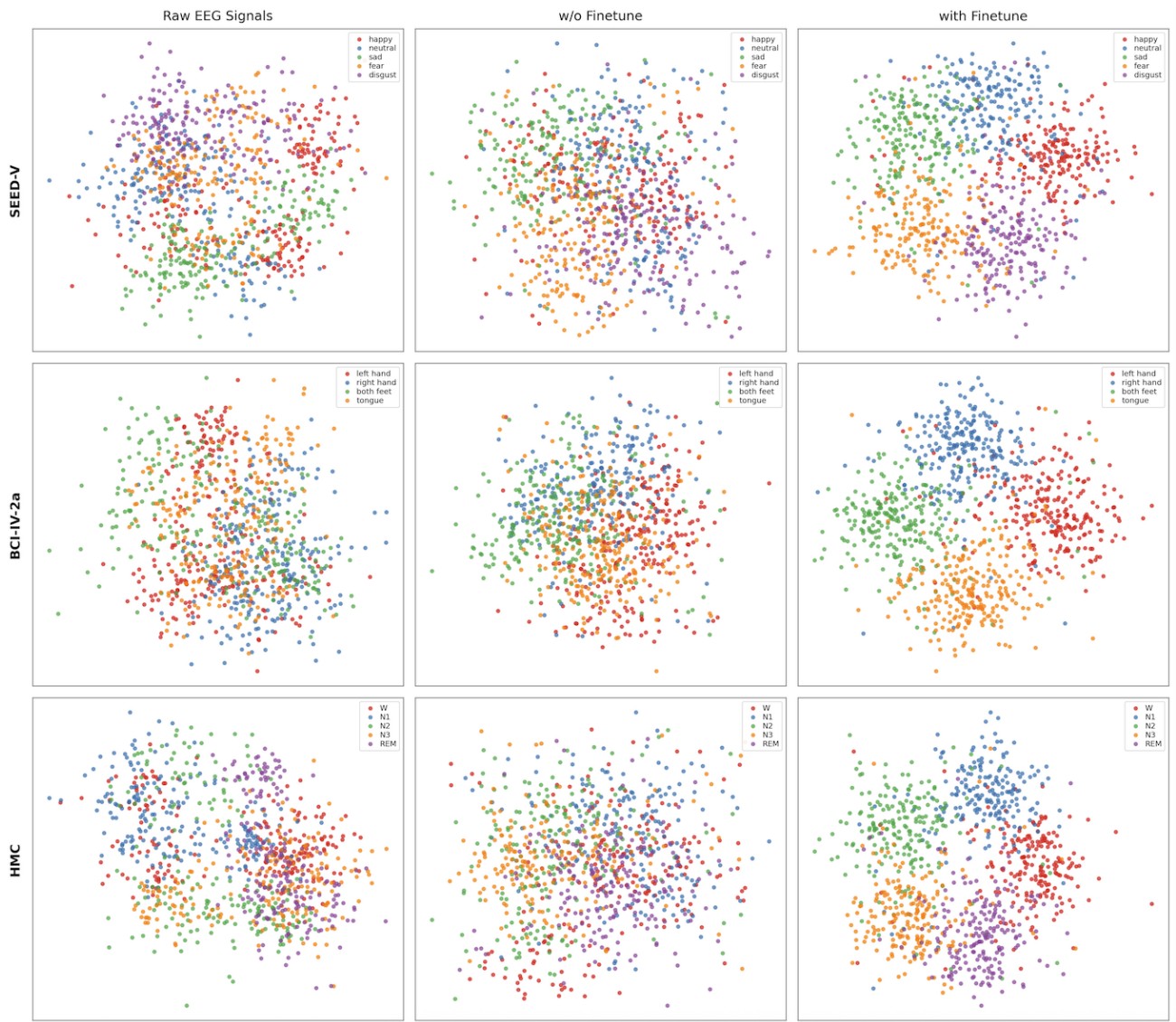

*Figure 6.* t-SNE visualization across tasks and adaptation stages. Rows show SEED-V, BCI-IV-2a, and HMC; columns show raw EEG, PATCHCODE without fine-tuning, and PATCHCODE with fine-tuning.

**Intra-class structure.** Beyond inter-class separation, we observe meaningful intra-class structure within each cluster. Points are not randomly distributed but show sub-clustering patterns, likely reflecting trial-specific variations in neural responses (e.g., different emotional intensities within the same emotion category, or varying motor imagery vividness). This hierarchical organization indicates that PATCHCODE preserves fine-grained temporal and spectral information while learning class-discriminative representations suitable for downstream classification.

