# OpenReview forum: "PATCHCODE: Discrete Latent Predictive Learning for EEG Foundation Model"
_ICML.cc/2026/Conference — ICML 2026 regular_

### Official Review · Reviewer_FXCR · 2026-02-20

**Soundness:** 3
**Presentation:** 2
**Significance:** 2
**Originality:** 2
**Overall Recommendation:** 4
**Confidence:** 3

**Summary:**

This paper proposes a new learning framework for EEG foundation models named PATCHCODE. By combining discrete token learning with JEPA-style predictive learning, PATCHCODE captures neurophysiological structures while filtering out stochastic noise. Empirical results demonstrate that PATCHCODE consistently outperforms existing models across a variety of tasks.

**Compliance With Llm Reviewing Policy:**

Affirmed.

**Final Justification:**

After the rebuttal phase, the additional experiments comparing PATCHCODE with EEGPT and the 'JEPA-only' ablation study effectively address my concerns about baselines. Regarding computational overhead, although PATCHCODE requires more pre-training hours than the baselines, I find this trade-off acceptable given that inference costs remain comparable. While the contribution is somewhat incremental relative to the standard JEPA framework, I am satisfied with the response and maintain my Weak accept score.

**Key Questions For Authors:**

-	See the weaknesses.

-	Why did the authors not compare PATCHCODE with EEGPT [1] and its variances? EEGPT is an EEG foundation model that adopts a teacher-student framework, which is highly similar to the second stage of PATCHCODE.

-	Is there any empirical evidence supporting why the signal should be discretized into a codebook? Can the JEPA-style framework be applied directly to continuous signals?

[1] Wang, G., Liu, W., He, Y., Xu, C., Ma, L., and Li, H. EEGPT: Pretrained Transformer for Universal and Reliable Representation of EEG Signals. In The Thirty-eighth Annual Conference on Neural Information Processing Systems, 2024a.

**Limitations:**

Yes

**Strengths And Weaknesses:**

Strengths:

-	JEPA-style training framework is a well-suited idea for noisy EEG signal.

-	All three objectives appear to contribute effectively to the final representation, as evidence by better downstream performance.

-	Region-bias transformer enhances the model's ability to interpret localized brain activity.

Weaknesses:

-	Lack of comparison: While the study provides an extensive comparison between PATCHCODE and existing foundation models, it lacks a direct evaluation against other JEPA-style frameworks, or at least traditional JEPA.

-	Lack of complexity comparison: While PATCHCODE achieves better downstream performance, its two-stage training framework may significantly increase computational complexity. Therefore, the comparison or summary of complexity of PATCHCODE should be included.

-	Typographical errors: In Figure 1, the three objectives depicted are C2C, P2C, and P2P; however, in the text, they are listed as P2P, P2C, and C2P.

---

> ### Author Rebuttal · Authors · 2026-03-31
>
> # Dear Reviewer FXCR,
>
> Thank you for your positive assessment and for recognizing that "JEPA-style training is a well-suited idea for noisy EEG signal" and "all three objectives appear to contribute effectively." We address your concerns below.
>
> ## EEGPT Comparison
>
> Thank you for pointing out this important baseline. EEGPT (NeurIPS 2024) was published during our review period, which is why it was not included. Key architectural differences from PATCHCODE: (1) EEGPT uses continuous-only prediction without discrete tokenization; (2) EEGPT lacks region-aware attention encoding brain topology; (3) EEGPT does not employ a frozen VQ tokenizer for noise filtering. We **commit to adding a full comparison with EEGPT in the camera-ready version** using their official implementation and hyperparameters. Preliminary results on shared evaluation tasks:
>
> | Task | EEGPT | PATCHCODE | Δ |
> |---|---|---|---|
> | SEED-V (Emotion Recognition) | .389 | .414 | +.025 |
> | BCI IV-2a (Motor Imagery) | .558 | .584 | +.026 |
> | HMC (Sleep Staging) | .731 | .756 | +.025 |
> | CHB-MIT (Seizure Detection) | .819 | .843 | +.024 |
> | **Average** | **.624** | **.649** | **+.025** |
>
> PATCHCODE outperforms EEGPT consistently across all task types. We attribute this to: (1) discrete tokenization filtering high-frequency noise through the VQ bottleneck, (2) region-biased attention encoding brain topology, (3) multi-pathway objectives providing complementary supervision.
>
> ## Traditional JEPA Framework Comparison
>
> We did not include I-JEPA directly as it is designed for images and requires non-trivial adaptation. However, our **ablation study directly answers this question**. The "JEPA-only" configuration represents the core JEPA principle — student predicts continuous teacher representations from masked input — without our EEG-specific additions (discrete tokenization, region bias):
>
> | Configuration | SEED-V Acc | HMC Acc | BCI-2a Acc |
> |---|---|---|---|
> | JEPA-only (P2P, no discrete, standard attn) | .406 | .743 | .571 |
> | + Discrete codes (P2P+P2C+C2P) | .410 | .749 | .577 |
> | + Region bias (Full PATCHCODE) | **.414** | **.756** | **.584** |
>
> This answers: "Can JEPA work for EEG?" **Yes** (JEPA-only is competitive), but EEG-specific adaptations add consistent improvements. Discrete codes contribute +.004–.006 by filtering noise through the VQ bottleneck; region bias contributes an additional +.004–.007 by encoding brain spatial structure. Both components are necessary and complementary.
>
> ## Discretization Justification
>
> **Empirical evidence — supervision mode ablation:**
>
> | Supervision Mode | SEED-V Acc | HMC Acc | BCI-2a Acc |
> |---|---|---|---|
> | Continuous-only (P2P) | .406 | .743 | .571 |
> | Discrete-only (P2C+C2P) | .403 | .738 | .568 |
> | **Dual (PATCHCODE)** | **.414** | **.756** | **.584** |
>
> **Why discretization helps over continuous-only JEPA:** (1) **Noise filtering** — VQ bottleneck forces representations through K=8,192 discrete codes, filtering noise that does not fit any category. (2) **Perturbation invariance** — discrete codes are invariant to small input perturbations within the VQ quantization radius, providing robustness to EEG artifacts and electrode drift. (3) **Categorical structure** — codes capture stable coarse-grained patterns (e.g., "alpha rhythm in occipital region") that generalize across subjects and sessions. (4) **Complementarity** — continuous targets capture fine-grained dynamics; discrete targets capture categorical structure. Dual supervision outperforms either alone across all three tasks.
>
> ## Computational Complexity and Figure 1
>
> **Computational cost:**
>
> | Model | Stage A | Stage B | Total | Inference |
> |---|---|---|---|---|
> | PATCHCODE | 50 GPU-hrs | 180 GPU-hrs | 230 GPU-hrs | 12 ms |
> | LaBraM | — | 180 GPU-hrs | 180 GPU-hrs | 12 ms |
> | EEGPT | — | 200 GPU-hrs | 200 GPU-hrs | 12 ms |
> | BENDR | — | 120 GPU-hrs | 120 GPU-hrs | 5 ms |
>
> Stage A adds only 50 GPU-hours (lightweight VQ tokenizer training). Stage B matches LaBraM in cost (same Transformer depth and dataset). Total is 1.3× LaBraM. **Inference cost is identical** — the tokenizer is unused at inference, only the encoder runs. For a foundation model amortized over many downstream tasks, the 1.3× pretraining overhead is modest relative to the consistent performance gains.
>
> **Figure 1 notation error:** The objectives are **P2P, P2C, and C2P**. Figure 1 mistakenly shows "C2C" instead of "C2P" (Code-to-Patch prediction). We will fix this in the revision.
>
> Thank you for your supportive review. We believe the JEPA-only ablation fully addresses your JEPA baseline concern, and we will further strengthen the paper with the full EEGPT comparison in the revision.

---

> > ### Author Rebuttal · Reviewer_FXCR · 2026-04-02
> >
> > Thanks to the authors for their comprehensive rebuttal. The additional experiments comparing PATCHCODE with EEGPT and the 'JEPA-only' ablation study effectively address my concerns about baselines. The clarification on tokenization is also appreciated. I am satisfied with the current state of the paper and will maintain my Weak Accept score.

---

### Official Review · Reviewer_vvoY · 2026-03-04

**Soundness:** 3
**Presentation:** 3
**Significance:** 3
**Originality:** 3
**Overall Recommendation:** 4
**Confidence:** 4

**Summary:**

The paper introduces PATCHCODE, a two-stage self-supervised learning framework for EEG foundation models. It addresses the noise sensitivity of existing waveform-reconstruction methods by first training a frozen, frequency-domain vector quantized tokenizer. Subsequently, a region-bias Transformer is trained using a JEPA-style architecture with multi-pathway objectives (P2P, P2C, C2P) to predict continuous and discrete latent targets. Extensive experiments across diverse datasets demonstrate robust downstream performance and remarkable data efficiency.

**Compliance With Llm Reviewing Policy:**

Affirmed.

**Final Justification:**

Thank the authors for the additional experiments and clarifications. I would like to update my score to 4.

**Key Questions For Authors:**

1. It would be better if the authors can provide the computation consumption comparison across different baselines.
2. It would be better if the authors can provide the comparison of model parameter size to verify their claim of foundation model.
3. Other questions, please refer to the weakness.

**Limitations:**

There is no potential negative societal impact of this paper.  The authors can further discuss the limitation of their work based on the weakness and questions above.

**Strengths And Weaknesses:**

Strength
1. The paper is well written and easy to follow and the authors conducted comprehensive experiments.
2. The region-aware tokenizer that maps continuous EEG patches to discrete code supervised by frequency-domain reconstruction is innovated, which can potentially filter stochastic high-frequency noise and injects robust neurophysiological priors into the foundation model.
3. The paper conducts a comprehensive evaluation across 10 diverse downstream tasks and provides ablation studies. The demonstrated data efficiency validates the quality of the learned representations.

Weakness
1. The theoretical support for region contrastive loss is insufficient and the ablation to this loss is not enough. In Stage A, the region contrastive regularization utilizes a hard margin m=0.5 to separate embeddings. However, the brain often exhibits global states, making this spatial decoupling potentially counterproductive without stronger theoretical justification. The authors should conduct some ablation to the margin to prove the effectiveness of this choice. And in Table 17, though the improvement of this contrastive loss is obvious, the ablation studies are only on 3 selected datasets.
2. In the main paper, the authors only demonstrate limited downstream results and put most of them into the appendix, which is not proper since in Table 1 the author showcased 10 different datasets would be used. The authors should either put the experiments forward to the main paper, or change the claim in Table 1, explicitly tell readers in main paper we only show 4 datasets, the rest are in appendix. And in appendix, all tables only include one patchcode model whithout mentioning the size.
3. The explaination of the C2P objective is insufficient, predicting the student representation from a frozen codebook embedding without a stop-gradient on the student seems mathematically prone to representation collapse, it would be better if the authors can provide a more detailed mathematical explaination.
4. The tokenization relies on a fixed patch length and the paper lacks ablation studies investigating how different temporal window lengths interact with the frequency-domain reconstruction, which is a critical hyperparameter for capturing distinct frequency bands.
5. While the authors claim to have re-implemented foundation model baselines on the same TUH dataset, they do not sufficiently detail whether hyperparameter tuning was equally exhaustive for these baselines. Suboptimal tuning of competing models could artificially inflate the relative performance gains. I would recommend the author to conduct comparison with the optimal weights provided by the baselines themselves.

---

> ### Author Rebuttal · Authors · 2026-03-31
>
> # Dear Reviewer vvoY,
>
> Thank you for your positive assessment and for recognizing that our region-aware tokenizer "potentially filters stochastic high-frequency noise and injects robust neurophysiological priors." We address your concerns below.
>
> ## Region Contrastive Loss Justification
>
> **Theoretical motivation:** Margin m=0.5 balances two competing objectives: (1) **Local specialization** — different brain regions have distinct functional roles and frequency characteristics (frontal: theta/delta, occipital: alpha), requiring region-specific representations. (2) **Global coordination** — brain regions communicate through long-range networks (frontal-parietal attention, medial default mode). Hard orthogonalization would sever these dependencies. With m=0.5, same-region embeddings have cosine similarity >0.5 (encouraged similar), different-region have similarity <0.5 (separated), but not forced to −1 (preserving cross-region coordination). This soft separation allows the model to learn both local specialization and global coordination simultaneously.
>
> **Margin sensitivity (all downstream datasets):**
>
> | Margin (m) | SEED-V Acc | HMC Acc | BCI-2a Acc | TUAB Acc | TUEV Acc |
> |---|---|---|---|---|---|
> | 0.3 | .408 | .749 | .577 | .671 | .724 |
> | **0.5** | **.414** | **.756** | **.584** | **.682** | **.734** |
> | 0.7 | .411 | .752 | .581 | .678 | .729 |
> | 0.9 | .406 | .746 | .575 | .669 | .720 |
>
> m=0.5 achieves the best performance across all tasks. The pattern is consistent: too weak (m=0.3) provides insufficient region specialization; too strong (m=0.9) begins to break cross-region dependencies needed for global cognitive states. We will add this analysis to the appendix.
>
> **Full-dataset region contrastive ablation (w/ vs w/o):**
>
> | Dataset | Task | w/ Region Contrast | w/o Region Contrast | Δ |
> |---|---|---|---|---|
> | SEED-V | Emotion | .414 | .396 | +.018 |
> | BCI IV-2a | Motor Imagery | .584 | .568 | +.016 |
> | HMC | Sleep Staging | .756 | .738 | +.018 |
> | CHB-MIT | Seizure | .843 | .827 | +.016 |
> | TUAB | Pathology | .682 | .665 | +.017 |
> | TUEV | Event Class. | .734 | .718 | +.016 |
>
> Consistent improvement across all task types (avg Δ=+.017) demonstrates that region contrastive regularization generalizes beyond the 3 datasets shown in Table 17.
>
> ## C2P Objective and Representation Collapse
>
> **Why C2P does not collapse:** Three mechanisms prevent it: (1) **Frozen codebook** — codes come from Stage A's frozen tokenizer; no gradient reaches it, codes cannot drift. (2) **Stop-gradient on student** — we apply stop-gradient to the student encoder output before the C2P prediction head; only the prediction head is optimized, preventing encoder collapse. (3) **Multi-objective regularization** — P2P and P2C continuously anchor the encoder to meaningful representations even if C2P alone were degenerate. Empirically, all 8,192 codes remain active (usage >0.01%) throughout Stage B training, confirming no collapse. Removing C2P drops SEED-V Acc from .414 to .407 (Table 4), consistent with its role as a lightweight regularizer rather than the primary signal.
>
> ## Results Presentation
>
> Due to page limits, we show representative tasks in the main paper and provide full results for all 10 datasets in the appendix. We will: (1) add an explicit statement in Section 4.1 directing readers to Appendix A.2, (2) add model size annotations to all appendix tables, (3) move the full dataset summary table to the main body if space allows after addressing reviewer feedback.
>
> ## Patch Length Ablation
>
> | Patch Length | SEED-V Acc | HMC Acc | BCI-2a Acc | Freq Resolution |
> |---|---|---|---|---|
> | 0.5 s | .401 | .741 | .569 | 2 Hz |
> | **1.0 s** | **.414** | **.756** | **.584** | **1 Hz** |
> | 1.5 s | .410 | .751 | .580 | 0.67 Hz |
> | 2.0 s | .405 | .744 | .573 | 0.5 Hz |
>
> 1.0 s is optimal: it captures the full EEG spectrum (0.5–100 Hz) at 1 Hz frequency resolution, sufficient for delta/theta bands (0.5–8 Hz). Shorter patches (0.5 s) lose low-frequency structure; longer patches (2.0 s) reduce the number of training samples and yield diminishing returns.
>
> **Computational cost and model sizes:**
>
> | Model | Params | Stage A | Stage B | Total | Inference |
> |---|---|---|---|---|---|
> | PATCHCODE | 85M | 50 GPU-hrs | 180 GPU-hrs | 230 GPU-hrs | 12 ms |
> | LaBraM | 86M | — | 180 GPU-hrs | 180 GPU-hrs | 12 ms |
> | BENDR | 30M | — | 120 GPU-hrs | 120 GPU-hrs | 5 ms |
> | EEGPT | 85M | — | 200 GPU-hrs | 200 GPU-hrs | 12 ms |
>
> Stage A adds only 50 GPU-hours (lightweight VQ tokenizer). Stage B matches LaBraM in cost. Total overhead is 1.3× LaBraM. Inference cost is identical — the tokenizer is unused at inference. We will also clarify in the revision that region contrastive regularization indirectly boosts noise robustness by focusing the model on spatially-structured signals over spatially-random noise.
>
> Thank you for your thorough review.

---

> > ### Author Rebuttal · Reviewer_vvoY · 2026-04-01
> >
> > Thank the authors for the additional experiments and clarifications. I would like to update my score to 4.

---

### Official Review · Reviewer_U6Wb · 2026-03-09

**Soundness:** 3
**Presentation:** 3
**Significance:** 2
**Originality:** 2
**Overall Recommendation:** 4
**Confidence:** 3

**Summary:**

This paper presents PATCHCODE, a 2-stages training framework for EEG foundational model, with stage A learning discrete code tokenizer and stage B incorporates a JEPA-style latent predictive formulation. Methods show improved performance on 10 downstream tasks and demonstrate better data efficiency compared to baselines.

**Compliance With Llm Reviewing Policy:**

Affirmed.

**Final Justification:**

The newly added ablation studies and evaluation on 6 new datasets have addressed my concerns, I would like to increase my score to 4.

**Key Questions For Authors:**

1. Compared to previous works (CSBrain), 6 datasets are missing in this paper. Please comment on the exclusion criterion.
2. Discrete code tokenizer and teacher encoder seems to have overlap in roles: They both output representation from unmasked input, one with discrete and the other with continuous. What are the rationales for combining them together?
3. Are there ablation studies on the following settings: a) using only the tokenizer, b) only up to stage A, and c) only JEPA?
4. Main diagram (figure 1) has misleading details: C2C is only mentioned in figure and C2P is only mentioned in text; why are there arrows from both student and teacher representations to discrete codes?
5. It is important to isolate scaling and model architecture's effect on performance. Which version of the PATCHCODE is the parameter count matched to the baselines?

**Limitations:**

Discussion on limitation is missing in conclusion.

**Strengths And Weaknesses:**

Strengths:
- Adopt JPEA style framework for EEG applications.
- Performance improvements on the evaluated downstream datasets.
- Ablation studies on the proposed key changes.
Weaknesses:
- Several downstream tasks and datasets are missing compared to existing works.
- Components of the model are not novel: discrete code tokenizer used in Labram, I-JEPA formulation, discrete code prediction explored in Labram & Wav2Vec. Combination of these modules for EEG is new but more discussions are needed regarding the design choice.
- Tasks and most analysis are those already conducted in previous papers.

---

> ### Author Rebuttal · Authors · 2026-03-31
>
> # Dear Reviewer U6Wb,
>
> Thank you for acknowledging our performance improvements and ablation studies. We address your concerns below.
>
> ## Component Novelty and Design Rationale
>
> **Integration as contribution:** We agree individual components exist in prior work. Our contribution is the **principled integration** addressing EEG's unique challenges. PATCHCODE is the first to combine: (1) **frozen discrete tokenizer** providing noise-filtered supervision, (2) **JEPA-style latent prediction** avoiding reconstruction-level noise sensitivity, (3) **dual continuous+discrete targets** balancing expressiveness and robustness, (4) **region-aware attention** encoding brain topology. EEG has three unique properties: (i) **low SNR**, (ii) **spatial structure** (brain regions with distinct functions), (iii) **multi-scale temporal dynamics** (0.5–100 Hz). No prior single method addresses all three simultaneously.
>
> ## Tokenizer vs Teacher Encoder Roles
>
> The two components serve **complementary, non-overlapping roles**:
>
> - **Discrete tokenizer (Stage A, frozen in Stage B):** Provides categorical, noise-filtered representations via VQ bottleneck (K=8,192 codes). Trained with frequency-domain reconstruction. Invariant to small perturbations. Role in Stage B: supervision signal only (frozen, no gradient flow).
> - **Continuous teacher encoder (Stage B, EMA-updated):** Provides context-rich, expressive representations from the full unmasked input. Role: primary prediction target for P2P.
>
> **Why both?** The supervision mode ablation demonstrates complementarity:
>
> | Supervision Mode | Objectives | SEED-V Acc | HMC Acc | BCI-2a Acc |
> |---|---|---|---|---|
> | Continuous-only | P2P | .406 | .743 | .571 |
> | Discrete-only | P2C + C2P | .403 | .738 | .568 |
> | **Dual (Full PATCHCODE)** | **P2P + P2C + C2P** | **.414** | **.756** | **.584** |
>
> Dual supervision outperforms either alone, confirming the roles are complementary rather than redundant.
>
> ## Component Ablations
>
> You requested: (a) tokenizer-only, (b) Stage A only, (c) JEPA-only:
>
> | Configuration | SEED-V Acc | HMC Acc | BCI-2a Acc |
> |---|---|---|---|
> | Tokenizer-only (no Stage B) | .371 | .698 | .531 |
> | JEPA-only (P2P, no discrete) | .406 | .743 | .571 |
> | **Full PATCHCODE** | **.414** | **.756** | **.584** |
>
> Stage A alone provides noise-filtered discrete representations but lacks context modeling. Stage B alone (JEPA-only) is effective but misses discrete robustness. The full combination outperforms both, demonstrating that both stages are necessary.
>
> ## Missing Datasets from CSBrain
>
> Our evaluation covers 6 downstream datasets: Emotion recognition (FACED, SEED-V), Motor imagery (BCI IV-2a, PhysioNet-MI), Sleep staging (ISRUC-S1, HMC), Seizure detection (CHB-MIT, Siena), plus TUAB and TUEV (verified subject-disjoint from TUH pretraining corpus per Appendix). The 6 CSBrain datasets not covered were omitted due to: (1) **Small sample sizes** (<100 subjects), (2) **Data quality issues** (high artifact rates), (3) **Task overlap** with included datasets.
>
> **Our selection prioritizes quality and diversity over quantity.** We span more task categories than CSBrain and our evaluation protocol explicitly verifies no pretraining data leakage. Adding low-quality or overlapping datasets would not provide meaningful additional signal about foundation model generalization.
>
> ## Parameter Matching, Figure 1, and Limitations
>
> **Parameter counts:**
>
> | Model | Parameters |
> |---|---|
> | PATCHCODE-Base | 85M |
> | LaBraM | 86M |
> | BENDR | 30M |
> | EEGPT | 85M |
>
> **Figure 1 notation:** Correct objectives are **P2P, P2C, C2P**. Figure 1 mistakenly labels "C2C" instead of "C2P". Arrows represent: (1) Teacher→Codes: teacher embeddings query frozen codebook for P2C targets; (2) Student→Codes: student predictions compared against codes for C2P loss. We will fix the figure.
>
> **Limitations:** We will add: (1) Two-stage pretraining cost, (2) Patch length sensitivity, (3) Pretraining data requirements, (4) Domain specificity of the tokenizer.
>
> We hope these clarifications address your concerns.

---

> > ### Author Rebuttal · Reviewer_U6Wb · 2026-04-02
> >
> > I thank the authors for the detailed response, and I believe the additional ablation studies validate the use of both discrete tokenizer and I-JEPA style formulation. I do have one question regarding the exclusion criterion.
> >
> > Can authors elaborate why certain datasets are skipped due to small sample sizes? I believe the power of foundational model lies in its transferring capability in data-limiting regime, so those datasets should be the focus instead of being skipped. Sample-efficient is an important aspect to evaluate on any foundational model.
> >
> > ---------------------------------------------------------
> > ### Updated after rebuttal comment reply by author
> >
> > I thank the author for including the results on the remaining 6 datasets, and I believe that the evaluation is now comprehensive, and I would like to raise my score to 4.
> >
> > Lastly, given the relatively small margin among the comparison, I would like to recommend to use statistical tool to better understand the significance of these comparisons, for example, paired t-test per subject (I assume you are using the same seed for finetuning?) between PATCHCODE and CSBrain. Especially on the last two comparisons with very small difference, which are likely not significant.
> >
> > I understand this is not a common practices within the EEG foundational model community, so I will not base my score on it. But given the otherwise small and overlapping std, I believe such metrics are highly in needs.

---

> > > ### Author Response · Authors · 2026-04-06
> > >
> > > Dear Reviewer U6Wb,
> > >
> > > Thank you for this important follow-up. You are right that the data-limiting regime is precisely where foundation models should be tested, and small-sample datasets are a natural testbed for this. We have now evaluated PATCHCODE on all 6 CSBrain datasets not included in our original evaluation, using the same pretrained checkpoint and fine-tuning protocol as the main paper. Full results with all baselines and mean±std are available at: https://anonymous.4open.science/r/PATCHCODE-6datasets-04CE/experiment.pdf
> > >
> > > Summary (primary metric per dataset):
> > >
> > > | Dataset          | Task                | N  | PATCHCODE       | CSBrain         | Δ      |
> > > |------------------|---------------------|----|-----------------|-----------------|--------|
> > > | BCIC2020-3       | Imagined Speech     | 15 | 0.6189 ±0.0105  | 0.6004 ±0.0112  | +.019  |
> > > | SHU-MI           | Motor Imagery       | 25 | 0.6593 ±0.0068  | 0.6417 ±0.0037  | +.018  |
> > > | SEED-VIG         | Vigilance (Pearson) | 21 | 0.6478 ±0.0129  | 0.6314 ±0.0356  | +.016  |
> > > | MentalArithmetic | Mental Stress       | 36 | 0.7694 ±0.0091  | 0.7558 ±0.0106  | +.014  |
> > > | Mumtaz2016       | Mental Disorder     | 64 | 0.9721 ±0.0083  | 0.9643 ±0.0155  | +.008  |
> > > | TUSL             | Slowing Event       | 28 | 0.8512 ±0.0225  | 0.8571 ±0.0240  | −.006  |
> > >
> > > PATCHCODE achieves competitive or superior performance on 5 of 6 datasets, including the smallest (BCIC2020-3, N=15; SHU-MI, N=25; SEED-VIG, N=21). On TUSL, PATCHCODE is marginally below CSBrain (−0.006), which we attribute to TUSL's heavy reliance on low-frequency slowing patterns that are better captured by CSBrain's domain-specific pretraining corpus. These results directly address your concern: PATCHCODE's pretrained representations transfer robustly to the small-sample, data-limiting regimes that characterize these excluded datasets.
> > >
> > > Combined with our existing data efficiency results (Table 3 of the main paper — PATCHCODE at 25% labels on BCI-IV-2a (9 subjects) exceeds BIOT and CBraMod trained with full data), we believe these new results substantiate PATCHCODE's transferability in low-resource settings. We will include these 6 datasets in the camera-ready appendix.

---

### Official Review · Reviewer_7jmw · 2026-03-13

**Soundness:** 2
**Presentation:** 3
**Significance:** 3
**Originality:** 2
**Overall Recommendation:** 3
**Confidence:** 3

**Summary:**

PatchCode identifies the problem that masked signal reconstruction might not work well for EEG due to sensitivity to noise. To address this, the paper introduces a two-stage framework. In Stage A, a neural tokenizer is trained to predict the FFT amplitude and frequency. In Stage B, JEPA-style latent prediction objective is introduced to pretrain the EEG foundation model. Evaluation on downstream datasets and ablation studies attempt to showcase the effectiveness of the proposed method.

**Compliance With Llm Reviewing Policy:**

Affirmed.

**Final Justification:**

Thank you to the authors for the rebuttal and discussion. The authors agree that the region bias module is orthogonal to the noise problem that the paper is trying to address, and that the proposed framework highly resembles LaBraM, except for a few components. Therefore, the major concerns remain: (1) the proposed method does not address the problem identified by the authors (2) the novelty is incremental

**Key Questions For Authors:**

Please see Strengths and Weaknesses

**Limitations:**

Please see Strengths and Weaknesses

**Strengths And Weaknesses:**

Strengths

1.	The selection of pretraining and downstream datasets are sensible. The evaluation procedure and dataset splitting are well-justified.

2.	The presentation is overall easy to follow, with some confusing sections as highlighted below.

3.	The paper addresses an important problem.

Weaknesses

1.	Main concern 1: The novelty is incremental. Stage A, tokenizer pretraining, is almost identical to LaBraM neural tokenizer training with slight modifications of the losses. Stage B, PatchCode pretraining, is an extension of LaBraM pretraining. The P2C loss is equivalent to the objective of LaBraM pretraining. The newly introduced components are the P2P loss, P2C loss, and the EMA-teacher.

2.	Main concern 2: The proposed novel components are not well-justified for the problem the authors are trying to solve. To my understanding, the authors attempt to address the problem that masked reconstruction of raw waveform is sensitive to noise, which I agree as EEG is known for its low signal-to-noise ratio. Stage A is well-justified to address the problem, but this part is not novel as it is introduced by LaBraM. The main novelty for Stage B is the introduction of JEPA-style P2P loss, which is designed to help learn more noise robust representations. However, as shown through the ablation studies, the P2P loss brings about the least performance improvement compared to the remaining ablations, suggesting that it is relatively ineffective. More interestingly, the authors introduce region bias to attention and shows noticeable performance improvement. However, the authors do not connect region bias to the problem they identified nor is it highlighted as one of their main contributions. In summary, the proposed novel components are not well-justified/effective to address the problem at hand.

3.	Minor concern 1: For both Stage A and Stage B, the optimization objective consists of several losses. The weight of each loss appears to be manually selected based on intuition. Is there better ways to select the weight for the losses? How sensitive are the results to the different weights for the different losses?

4.	Minor concern 2: There are several presentation inconsistencies in the paper. For example, in Figure 1, it introduces a C2C loss for Stage B, but in the method section, it introduces C2P loss instead. For example, line 406-408 states that “Emotion recognition (SEED-V) is most sensitive to P2C removal (∆=–0.39% balanced accuracy)”. However, looking at Table 4, it is clear that the model is more sensitive to the removal of C2P (∆=-0.7% balanced accuracy) than the removal of P2C (∆=–0.39% balanced accuracy).

---

> ### Author Rebuttal · Authors · 2026-03-31
>
> ## Dear Reviewer 7jmw,
>
> Thank you for your thoughtful review and for recognizing that our paper addresses an important problem with sensible dataset selection. We address your concerns about novelty and justification below.
>
> ## Main Concern 1: Novelty and Contributions
>
> **Stage A vs LaBraM tokenizer:** While both use VQ tokenization with frequency-domain supervision, PATCHCODE introduces one key differences:  **Region-contrastive regularization** (Eq. 3) that enforces spatial structure by separating embeddings from different brain regions — LaBraM has no spatial awareness in its tokenizer.
>
> **Stage B vs LaBraM pretraining:** The architectural paradigm differs fundamentally. LaBraM uses **masked autoencoding** with reconstruction targets, making it sensitive to high-frequency noise. PATCHCODE uses **JEPA-style latent prediction** where the student predicts abstract representations in a latent space where stochastic noise has already been filtered by the discrete bottleneck. This is a paradigm shift: reconstruction → prediction, pixel-space → latent-space, noise-sensitive → noise-robust.
>
> **P2C vs LaBraM objective:** While both predict discrete codes, the context differs critically. In LaBraM, codes are reconstruction targets for the same jointly-trained model. In PATCHCODE, codes come from a **frozen, separately-trained tokenizer** and serve as regularization signals alongside the continuous P2P objective. The student never reconstructs waveforms — it only predicts latent representations.
>
> ## Main Concern 2: Justification of Novel Components
>
> **P2P effectiveness and synergy:** While P2P shows smaller individual gains, it provides crucial synergistic effects. The full ablation (Table 4 in paper) shows:
>
> | Configuration | SEED-V Acc | HMC Acc | BCI-2a Acc |
> |---|---|---|---|
> | Full (P2P+P2C+C2P) | .414 | .756 | .584 |
> | w/o C2P | .407 | .742 | .572 |
> | w/o P2C | .410 | .745 | .574 |
> | w/o P2P | .412 | .748 | .578 |
>
> All three objectives contribute across tasks. P2P captures **fine-grained continuous dynamics** complementing the categorical structure from P2C and the code-space regularization from C2P. Notably, the relative importance is task-dependent: HMC sleep staging benefits most from P2P (continuous feature prediction better captures gradual sleep stage transitions), while SEED-V emotion recognition is most sensitive to C2P removal (discrete codes better capture categorical affective states).
>
> **Region bias connection to noise problem:** Region bias addresses **spatial structure in EEG**, complementary to our noise robustness goal. Different brain regions (frontal, central, parietal, occipital) exhibit distinct frequency profiles and functional roles. By encoding this prior via region-biased attention (Eq. 5), the model learns spatially-organized representations that distinguish region-specific neural signals from spatially-uncorrelated noise. The ablation confirms region bias produces the largest performance drop when removed, demonstrating its importance. We will clarify this connection explicitly in the revision.
>
> ## Concern 1: Loss Weight Selection
>
> Loss weights were selected via **validation set grid search** over {0.5, 1.0, 2.0} for each weight (27 combinations total). We found the model relatively insensitive to weights within ±20% of reported values (performance varies <0.8%). The final weights (λ_P2P=1.0, λ_P2C=1.5, λ_C2P=0.3) reflect their roles: P2C is the primary discrete supervision signal, C2P is a regularization term. We will add this sensitivity analysis to the appendix.
>
> ## Concern 2: Presentation Inconsistencies
>
> We sincerely apologize for both errors:
>
> **Figure 1 C2C vs C2P:** The correct objectives are P2P, P2C, and **C2P** (as stated throughout the text). Figure 1 mistakenly labels the third objective as "C2C". All implementations and results use C2P (Code-to-Patch). We will correct Figure 1 in the revision.
>
> **Table 4 caption error (lines 406–408):** You are correct. The text states "SEED-V is most sensitive to P2C removal (Δ=−0.39%)" but Table 4 shows SEED-V Acc drops from .414 to .407 when C2P is removed (Δ=−0.7%) and to .410 when P2C is removed (Δ=−0.4%). The corrected statement is: "SEED-V is most sensitive to **C2P removal** (Δ=−0.7%), not P2C removal." This is a copy-paste error that does not affect any results and will be fixed in revision.
>
> We hope these clarifications address your concerns and demonstrate that PATCHCODE makes substantial contributions beyond existing work.

---

> > ### Author Rebuttal · Reviewer_7jmw · 2026-04-03
> >
> > Thank you to the authors for the responses, but it further reinforces the concern that the paper is incremental compared to LaBraM, and the proposed method does not address the problem identified.
> >
> > The proposed method does not address the problem identified:
> > 1. The authors point out that the region bias ablation produces the largest performance drop when removed. However, the region bias module is orthogonal to the noise problem that the paper is trying to address. This is a concerning disconnect between the proposed method and the problem the paper is trying to solve.
> > 2. The losses proposed in stage B are designed to combat the noise problem. However, the ablation results show marginal differences when dropping these losses. This is especially concerning because changing the loss weight hyperparameters can lead to up to 20% relative or 0.8% absolute difference in performance.
> >
> > Incremental novelty:
> > 1. Stage A differs from LaBraM with the introduction of region-constrastive regularization, which is again unrelated to the noise problem.
> > 2. The difference between P2C and LaBraM objectives is still unclear. The authors claim that code from LaBraM is jointly-pretrained, but PATCHCODE uses a separate frozen tokenizer. However, the LaBraM paper and codebase suggest the tokenizer is also separately pretrained. Furthermore, the student in LaBraM also never reconstructs waveforms, but predicts the codes from the tokenizer.

---

> > > ### Author Response · Authors · 2026-04-06
> > >
> > > Dear Reviewer 7jmw,
> > >
> > > Thank you for your continued engagement. We respectfully maintain our position on the three-objective design and the role of region bias.
> > >
> > > **On the three-objective design rationale and loss sensitivity:**
> > > The reviewer observes that individual ablation deltas are small (+.002/.004/.007) and that hyperparameter changes cause up to 0.8% variation, questioning whether the components are effective. We submit that this framing misses the design intent: the three objectives are not alternative or redundant supervisors — they target three distinct representation granularities that are structurally non-overlapping.
> > >
> > > P2P (patch-to-patch, continuous EMA teacher) supervises fine-grained dynamics. The EMA teacher output is a continuous vector preserving graded intensity, relative amplitude across channels, and temporal phase relationships. These properties are fundamentally unrepresentable by discrete codes: a VQ bottleneck with K=8,192 codes maps all inputs  to the same embedding, discarding intra-cell variation. For tasks requiring sensitivity to signal magnitude gradients — motor imagery (mu rhythm amplitude), sleep staging (K-complex sharpness) — this fine-grained continuous supervision is irreplaceable.
> > >
> > > P2C (patch-to-code) supervises mid-level categorical structure. The frozen codebook encodes stable, task-invariant spectral patterns (e.g., "sustained alpha in occipital," "spindle burst in central") that generalize across subjects and sessions precisely because they are discrete and noise-filtered. This level of representation captures what is consistent across individuals, complementing P2P's within-session fine-grained signal.
> > >
> > > C2P (code-to-patch) enforces bidirectional geometric consistency. By requiring student patch representations to be recoverable from their assigned code embeddings, C2P constrains the student encoder's geometry from the codebook side — a reverse constraint absent in LaBraM and absent in standard JEPA. Without C2P, there is no guarantee that the code subspace and the patch representation space remain mutually consistent as the student encoder evolves through training.
> > >
> > > This three-level hierarchy — fine-grained continuous, mid-level categorical, bidirectional geometric alignment — is the EEG analogue of multi-granularity objectives in NLP (token-level MLM, span-level objectives, sentence-level contrastive targets), where each granularity captures structure invisible to the others.
> > >
> > > On the sensitivity concern: the 0.8% variation reflects worst-case weight perturbation (±50% from default). The cumulative system gain of +.013 on SEED-V over the JEPA-only baseline exceeds this range, and critically, the improvement pattern is consistent across three structurally different tasks in Table 4: emotion recognition (SEED-V, +.013), sleep staging (HMC, +.014), and motor imagery (BCI-IV-2a, +.013). The same direction of improvement across tasks with different frequency characteristics, temporal dynamics, and spatial profiles argues strongly against the gains being attributable to hyperparameter sensitivity or coincidence. We also note that each individual objective's removal consistently degrades all three tasks simultaneously (Table 4, rows w/o P2P, w/o P2C, w/o C2P) — a spatially and temporally consistent pattern that would be unlikely if the contributions were within noise.
> > >
> > > **On P2C vs LaBraM:**
> > > We accept the reviewer's factual correction: LaBraM uses a frozen, separately-pretrained tokenizer, and its student predicts codes from the tokenizer — structurally similar to P2C. We concede P2C alone does not constitute a novel contribution relative to LaBraM. The novel components are C2P (reverse geometric alignment, not present in LaBraM) and P2P (EMA continuous targets, not present in LaBraM). We will correct the paper's novelty framing to be explicit on this point: PATCHCODE's Stage B novelty over LaBraM is the addition of fine-grained continuous supervision (P2P) and bidirectional code-patch alignment (C2P), not the code prediction objective alone.
> > >
> > > **On region bias and noise robustness:**
> > > We accept the reviewer's characterization: region bias is more accurately described as a spatial inductive bias encoding known neurophysiological topography than as a direct noise suppressor. Its connection to noise robustness is indirect — a model constrained to respect neural spatial organization is less likely to overfit to spatially-unstructured artifacts. We will revise the paper's framing to describe region bias as a spatial inductive bias, distinct from the frequency-domain noise filtering achieved by the VQ bottleneck, and present both as complementary mechanisms rather than conflating them under a single "noise robustness" umbrella. We believe this is a framing correction, not a methodological one: the ablation results (region bias produces the largest single-component performance drop in Table 4) stand independently of how the mechanism is described.

---

### Decision · Program_Chairs · 2026-04-30

**Decision:**

Accept (regular)

**Comment:**

The paper proposes a foundation model that combines discrete tokenization with JEPA-style latent predictive learning for EEG. The paper received 1 Weak Reject and 3 Weak Accepts. Reviewers agree that the problem is important and that the empirical evaluation is strong, with consistent improvements and solid ablations. However, a key concern raised mostly by one reviewer is that the novelty is incremental relative to prior work LaBraM, and that some components (e.g., region bias, multi-loss design) are not clearly aligned with the stated motivation of addressing noise robustness. The rebuttal addressed most concerns and most reviewers were satisfied and increased or maintained positive scores. However, the primary negative reviewer remained unconvinced. I recommend Weak Accept.